# Exploring Invariance in Images through One-way Wave Equations

**Yinpeng Chen** [† 1]  **Dongdong Chen** [2]  **Xiyang Dai** [2]  **Mengchen Liu** [† 3]  **Yinan Feng** [4]  **Youzuo Lin** [4]  **Lu Yuan** [† 3]
**Zicheng Liu** [† 5]

## Abstract

In this paper, we empirically demonstrate that natural images can be reconstructed with high fidelity from compressed representations using a simple first-order norm-plus-linear autoregressive (FINOLA) process—without relying on explicit positional information. Through systematic analysis, we observe that the learned coefficient matrices ($\mathbf{A}$ and $\mathbf{B}$) in FINOLA are typically invertible, and their product, $\mathbf{A}\mathbf{B}^{-1}$, is diagonalizable across training runs. This structure enables a striking interpretation: FINOLA's latent dynamics resemble a system of one-way wave equations evolving in a compressed latent space. Under this framework, each image corresponds to a unique solution of these equations. This offers a new perspective on image invariance, suggesting that the underlying structure of images may be governed by simple, invariant dynamic laws. Our findings shed light on a novel avenue for understanding and modeling visual data through the lens of latent-space dynamics and wave propagation.

## 1. Introduction

**Motivation:** Autoregressive language models, such as GPT (Radford et al., 2018; 2019; Brown et al., 2020), have achieved remarkable success in Natural Language Processing (NLP) by predicting each token based on its preceding context. This paradigm has also influenced Computer Vision, inspiring models like iGPT (Chen et al., 2020a) for unsupervised representation learning, PixelCNN (van den Oord et al., 2016a; Salimans et al., 2017) for autoregressive image generation, and DALL·E (Ramesh et al., 2021) for text-to-image synthesis. These approaches typically rely on capturing high-order dependencies among multiple tokens,

---

† Work done while working at Microsoft. [1]Google DeepMind [2]Microsoft [3]Meta [4]University of North Carolina at Chapel Hill [5]AMD. Correspondence to: Yinpeng Chen <yinpengc@google.com>.

*Proceedings of the $42^{nd}$ International Conference on Machine Learning*, Vancouver, Canada. PMLR 267, 2025. Copyright 2025 by the author(s).

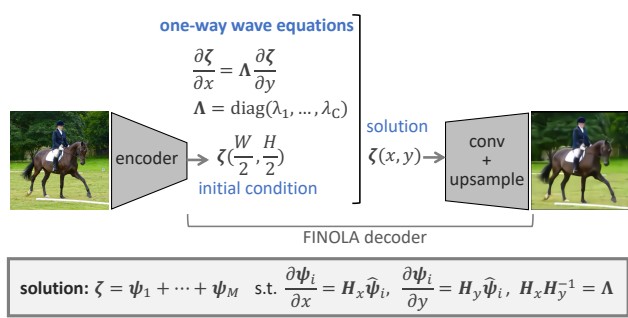

Figure 1: **Exploring invariance through one-way wave equations**. Our empirical findings suggest that images may share a set of one-way wave equations $\frac{\partial \boldsymbol{\zeta}}{\partial x} = \boldsymbol{\Lambda}\frac{\partial \boldsymbol{\zeta}}{\partial y}$ (or transportation equations). Each image corresponds (to a good approximation) to a unique solution with an initial condition $\boldsymbol{\zeta}(\frac{W}{2}, \frac{H}{2})$ derived from the original image. The solution $\boldsymbol{\zeta}(x, y)$ is a feature map (with resolutions of $\frac{1}{4}$ or $\frac{1}{2}$ or full resolution of the original image) facilitates image reconstruction using a few upsampling and convolutional layers. The wave speeds, $\lambda_1, \ldots, \lambda_C$, are latent and learnable.

often through deep Transformer architectures. Motivated by their success, we ask: *can autoregressive modeling be simplified to a first-order process, while still retaining its expressive power?*

**First-order norm+linear autoregression (FINOLA):** We propose a simplified *first-order* autoregressive process for image reconstruction, which we term FINOLA (First-Order Norm+Linear Autoregression). Our key insight is that, with appropriate encoding, images can be autoregressed linearly after normalization. As illustrated in Figure 2, the process begins by encoding an image into a compact vector $\boldsymbol{q} \in \mathbb{R}^C$. This vector is placed at the center of a latent feature map $\boldsymbol{z} \in \mathbb{R}^{W \times H \times C}$, i.e., $\boldsymbol{z}(\frac{W}{2}, \frac{H}{2}) = \boldsymbol{q}$. We then recursively propagate values across the $x$ and $y$ axes using: $\Delta_x \boldsymbol{z} = \boldsymbol{z}(x+1, y) - \boldsymbol{z}(x, y) = \boldsymbol{A}\hat{\boldsymbol{z}}(x, y)$, and $\Delta_y \boldsymbol{z} = \boldsymbol{z}(x, y+1) - \boldsymbol{z}(x, y) = \boldsymbol{B}\hat{\boldsymbol{z}}(x, y)$, where $\boldsymbol{A}, \boldsymbol{B} \in \mathbb{R}^{C \times C}$ are learnable matrices, and $\hat{\boldsymbol{z}}(x, y)$ is the channel-wise normalized version of $\boldsymbol{z}(x, y)$: $\hat{\boldsymbol{z}}(x, y) = \frac{\boldsymbol{z}(x,y) - \mu_{\boldsymbol{z}}}{\sigma_{\boldsymbol{z}}}$, $\mu_{\boldsymbol{z}} = \frac{1}{C}\sum_k z_k(x, y)$, $\sigma_{\boldsymbol{z}} = \sqrt{\sum_k (z_k - \mu_{\boldsymbol{z}})^2 / C}$. FINOLA can generate high-resolution feature maps (e.g., $\frac{1}{4}$, $\frac{1}{2}$, or full resolution of the original image). The final image is reconstructed using a lightweight decoder composed of a

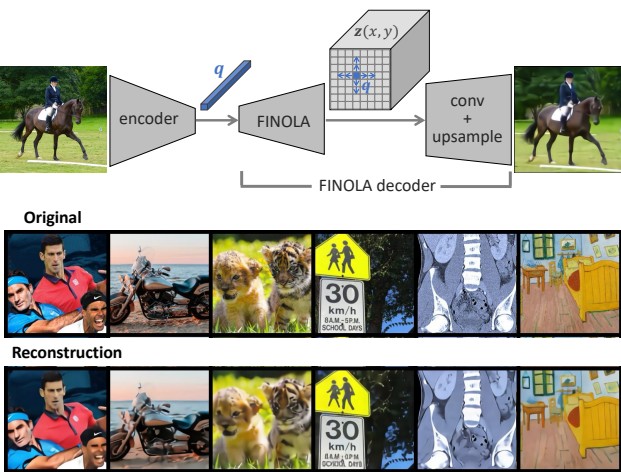

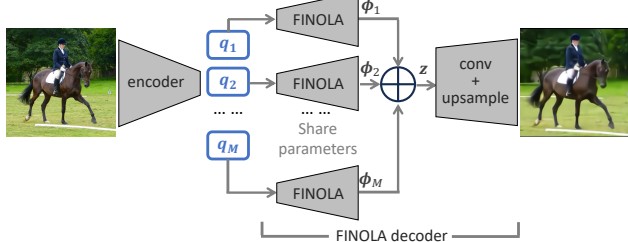

Figure 3: **Multi-path FINOLA:** The input image is encoded into $M$ vectors $\boldsymbol{q}_1, \ldots, \boldsymbol{q}_M$. Then the shared FINOLA is applied on each $\boldsymbol{q}_i$ to generate feature maps $\boldsymbol{\phi}_i(x, y)$, which are aggregated ($\boldsymbol{z} = \sum_i \boldsymbol{\phi}_i$) to pass through upsampling and convolution layers to reconstruct image pixels.

Figure 2: **FINOLA for image reconstruction.** Each image is firstly encoded into a single vector $\boldsymbol{q}$. Then, FINOLA is applied to $\boldsymbol{q}$ to iteratively generate the feature map $\boldsymbol{z}(x, y)$ through a first-order norm+linear autoregression. Finally, a few upsampling and convolutional layers are used to reconstruct image pixels. Best viewed in color.

few upsampling and convolutional layers.

**Reconstruction based on local structure alone:** The learned coefficient matrices $\boldsymbol{A}$ and $\boldsymbol{B}$ are invariant not only across spatial positions $(x, y)$ within a single image but also across different images. These matrices encode a consistent local relationship between a feature vector $\boldsymbol{z}(x, y)$ and its rate of change $\Delta \boldsymbol{z}(x, y)$, enabling FINOLA to reconstruct an entire image from a single central vector. Remarkably, this reconstruction relies solely on local propagation rules—without requiring explicit position encoding or global structure. On ImageNet (Deng et al., 2009) ($256 \times 256$ resolution), FINOLA achieves promising reconstruction performance. With a latent dimension of $C = 128$, we obtain a PSNR of 23.2 on the validation set. Increasing the dimension to $C = 2048$ improves the PSNR to 29.1. Compared to traditional encoding/decoding methods under the same latent size, FINOLA outperforms the discrete cosine transform (DCT), discrete wavelet transform (DWT), and convolutional autoencoders (AEs).

**Empirical properties of the coefficient matrices $\boldsymbol{A}$ and $\boldsymbol{B}$:** Our empirical analysis reveals that the learned FINOLA matrices $\boldsymbol{A}$ and $\boldsymbol{B}$ are typically invertible and that their product, $\boldsymbol{A}\boldsymbol{B}^{-1}$, is diagonalizable across multiple training runs. That is, $\boldsymbol{A}\boldsymbol{B}^{-1} = \boldsymbol{V}\boldsymbol{\Lambda}\boldsymbol{V}^{-1}$, where $\boldsymbol{\Lambda}$ is a diagonal matrix and $\boldsymbol{V}$ contains the eigenvectors.

**Interpretation via one-way wave equations:** Given the invertibility of $\boldsymbol{A}$ and $\boldsymbol{B}$, FINOLA's dynamics can be reformulated as a linear difference equation: $\Delta_x \boldsymbol{z} = \boldsymbol{A}\boldsymbol{B}^{-1}\Delta_y \boldsymbol{z}$, where $\Delta_x \boldsymbol{z}$ and $\Delta_y \boldsymbol{z}$ denote first-order differences along the $x$- and $y$-axes, respectively. Since $\boldsymbol{A}\boldsymbol{B}^{-1}$ is diagonalizable, we can express this relation in a transformed feature space $\boldsymbol{\zeta}(x, y) = \boldsymbol{V}^{-1}\boldsymbol{z}(x, y)$ as: $\Delta_x \boldsymbol{\zeta} = \boldsymbol{\Lambda}\Delta_y \boldsymbol{\zeta}$. In this space, the channels $\zeta_k$ are decorrelated, and each follows a separable form: $\Delta_x \zeta_k = \lambda_k \Delta_y \zeta_k$, which can be interpreted as a finite-difference approximation of a one-way wave equation: $\frac{\partial \zeta_k}{\partial x} = \lambda_k \frac{\partial \zeta_k}{\partial y}$. As illustrated in Figure 1, each image corresponds to a unique solution of this system of equations, fully determined by its initial condition at the center of the feature map: $\boldsymbol{\zeta}(\frac{W}{2}, \frac{H}{2}) = \boldsymbol{V}^{-1}\boldsymbol{q}$. This observation leads to a key insight:

> *Natural images may share a common system of one-way wave equations in latent space; each image is uniquely characterized by its initial condition, from which its full structure can be reconstructed.*

**FINOLA for self-supervised pre-training:** Beyond reconstruction, FINOLA can be adapted for self-supervised pre-training. Specifically, applying FINOLA to a single unmasked quadrant block and training it to predict the surrounding masked regions yields competitive performance compared to established approaches such as MAE (He et al., 2021) and SimMIM (Xie et al., 2022)—even when using lightweight architectures like Mobile-Former (Chen et al., 2022). Comparing encoders trained with and without masking reveals a trade-off: while masked prediction reduces reconstruction fidelity, it significantly enhances the learned semantic representations. Interestingly, we observe that masking leads to an increase in Gaussian curvature on the surfaces of critical feature maps, suggesting that the model develops more abstract and topologically complex representations under masking constraints.

**Acknowledging theoretical limitations:** While our empirical results strongly support the wave equation interpretation of FINOLA, we acknowledge that this perspective currently

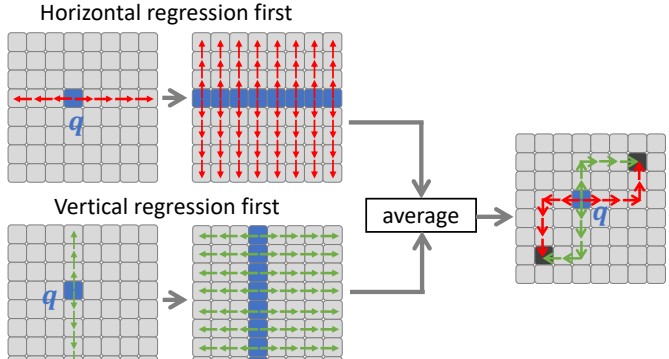

Horizontal regression first

Vertical regression first

average

**Regular vs. Parallel autoregression**. End-to-end runtime of encoder/decoder for a 256x256 image on a MacBook Air with an Apple M2 CPU is reported.

| Autoregression | Feature Resolution | |
| --- | --- | --- |
| | 16x16 | 64x64 |
| Regular | 1.7s | 7.7s |
| Parallel | **1.2s** | **2.6s** |

Figure 4: **Parallel implementation of FINOLA:** Horizontal and vertical regressions are separated. The *top* approach performs horizontal regression first, enabling parallel vertical regression. Similarly, the *bottom* approach starts with vertical regression, enabling parallel horizontal regression. The results of these approaches are averaged, corresponding to the two autoregression paths from the initial position marked by $q$. Best viewed in color.

lacks a rigorous theoretical foundation. We hope that our findings inspire future work to establish a deeper theoretical understanding of the dynamics uncovered in this study.

## 2. First-order Norm+Linear Autoregression

In this section, we introduce a first-order norm+linear autoregressive process in the latent space, known as FINOLA, which is able to reconstruct the entire image from a single vector at the center. It unveils a *position-invariant* and *image-invariant* relationship between the feature values $z(x, y)$ (at any $(x, y)$ position for any image) and its spatial rate of changes $\Delta_x z(x, y)$ and $\Delta_y z(x, y)$.

**FINOLA:** FINOLA is a first-order norm+linear autoregressive process that generates a $W \times H$ feature map $z(x, y)$ by predicting each position using only its immediate previous neighbor. As depicted in Figure 2, it places a single embedding $q$ (generated by an encoder) at the center, i.e., $z(\frac{W}{2}, \frac{H}{2}) = q$, and recursively regresses the entire feature map using the following equations:

$$z(x + 1, y) = z(x, y) + A\hat{z}(x, y) \quad \hat{z}(x, y) = \frac{z(x, y) - \mu_z}{\sigma_z}$$
$$z(x, y + 1) = z(x, y) + B\hat{z}(x, y) \quad \tag{1}$$

The matrices $A$ and $B$ are learnable with dimensions $C \times C$. $\hat{z}(x, y)$ is the normalized $z(x, y)$ over $C$ channels at position $(x, y)$: the mean $\mu_z = \frac{1}{C} \sum_k z_k(x, y)$ and the standard deviation $\sigma_z = \sqrt{\sum_k (z_k - \mu_z)^2 / C}$ are computed per position $(x, y)$ over $C$ channels. Due to the normalization, this process is a ***first-order non-linear*** process.

Eq. 1 provides a solution to predict towards the right and down (assuming the $y$ axis points down). For predicting towards the left and up (with negative values of offset), we introduce two additional learnable matrices, $A_-$ and $B_-$,

to perform predictions in the same manner as for right and down directions. Specifically, prediction toward the left is expressed as $z(x - 1, y) = z(x, y) + A_- \hat{z}(x, y)$. For brevity, we omit $A_-$ and $B_-$ in the rest of the paper.

Finally, image pixels are reconstructed by passing the feature map $z$ through upsampling and convolutional layers, as depicted in Figure 2. Remarkably, FINOLA exhibits the ability to generate the feature map $z$ at high resolutions, including $\frac{1}{4}$, $\frac{1}{2}$, or full resolution of the original image. In the most extreme scenario, where the feature map matches the resolution of the original image, merely three $3 \times 3$ convolutional layers are required to generate the image pixels.

The entire FINOLA framework, comprising the encoder, FINOLA, and the subsequent upsampling/convolutional layers, can be trained in an end-to-end manner. This is achieved by minimizing the $L_2$ distance between the original and the reconstructed images as the training loss.

**Position and image invariance:** Note that the matrices $A$ and $B$, once learned from data, remain invariant not only across spatial positions $(x, y)$ per image but also across images. They capture the local relation between the feature values $z(x, y)$ and their spatial derivatives $(\Delta_x z, \Delta_y z)$.

**Interpretation of matrices $A$ and $B$:** Imagine points in the latent feature map connected by "springs". Matrices $A$ and $B$ define the stiffness of these horizontal and vertical springs, while their eigenvectors $V_A$ and $V_B$ indicate their directions. We confirm $A$ and $B$ full rank and diagonalizable across multiple training runs. Diagonalizing $A$ and $B$ simplifies this, projecting latent space so each dimension (eigenvector) is an independent spring with stiffness (eigenvalue). The projected horizontal change $V_A^{-1} \Delta_x z$ and vertical change $V_B^{-1} \Delta_y z$ become scaling operations, with eigenvalues from $\Lambda_A$ and $\Lambda_B$ determining scaling

strength along each eigenvector. In essence, $\boldsymbol{A}$ and $\boldsymbol{B}$ encode directional "stretching/compressing" factors governing local feature map changes, with eigenvalues representing the strength of these factors.

**Parallel implementation:** Autoregression can be computationally intensive due to its sequential nature. FINOLA mitigates this by capitalizing on the independence of the $x$ and $y$ axes, enabling parallel execution, significantly boosting efficiency. As shown in Figure 4, performing horizontal regression first allows for parallel execution of subsequent vertical regression, and vice versa. In practice, both approaches (horizontal first and vertical first) are combined by averaging their results. The prediction at each position represents the average of the two autoregression paths originating from the initial position, marked as $\boldsymbol{q}$. Figure 4 (table on the right) demonstrates the superior speed of the parallel implementation, compared to the regular AR setting. It achieves a 30% speedup at a resolution of $16\times16$ and a threefold increase in speed at a higher resolution of $64\times64$.

**Importance of Norm+Linear:** In Section 4.1, experiments support the significance of *Norm+Linear* by showing that (a) simpler processes such as repetition or linear without normalization lead to significant degradation, (b) per-sample normalization is crucial, as seen in poor performance of Batch-Norm during validation, and (c) the gain from more complex non-linear models (e.g. MLP) is negligible.

## 3. Interpretation via One-way Wave Equations

In this section, we interpret FINOLA from the a perspective of one-way wave equations, empirically offering a deeper insight into the inherent nature of images.

**Linear partial difference equations:** Let's denote the spatial increments of the feature $\boldsymbol{z}$ along $x$ and $y$ axes as $\Delta_x \boldsymbol{z} = \boldsymbol{z}(x+1,y) - \boldsymbol{z}(x,y)$ and $\Delta_y \boldsymbol{z} = \boldsymbol{z}(x,y+1) - \boldsymbol{z}(x,y)$, respectively. Then, FINOLA (Eq. 1) can be expressed as linear partial difference equations:

$$\Delta_x \boldsymbol{z} = \boldsymbol{A}\boldsymbol{B}^{-1}\Delta_y \boldsymbol{z} = \boldsymbol{Q}\Delta_y \boldsymbol{z} \ \ s.t. \ \boldsymbol{Q} = \boldsymbol{A}\boldsymbol{B}^{-1}. \quad (2)$$

Here, the horizontal change $\Delta_x \boldsymbol{z}$ exhibits a linear correlation with its vertical counterpart $\Delta_y \boldsymbol{z}$. When the matrix $\boldsymbol{B}$ is invertible, FINOLA stands as a special solution to this equation, given that $\Delta_x \boldsymbol{z}$ and $\Delta_y \boldsymbol{z}$ not only exhibit linear correlation but are also linearly correlated with the normalization of the current feature values $\hat{\boldsymbol{z}}$ (referred to as the *FINOLA constraint*). It's noteworthy that the learned matrices $\boldsymbol{A}$ and $\boldsymbol{B}$ are empirically found to be invertible across multiple training runs with various dimensions, ranging from $128\times128$ to $4096\times4096$.

**Relaxing the FINOLA constraint through FINOLA series:** FINOLA represents a specific solution to Eq. 2, but it may not be the optimal one. We have discovered that a more

optimal solution can be attained by relaxing the FINOLA constraint ($\Delta_x \boldsymbol{z} = \boldsymbol{A}\hat{\boldsymbol{z}}, \Delta_y \boldsymbol{z} = \boldsymbol{B}\hat{\boldsymbol{z}}$) through aggregating a series of FINOLA solutions:

$$\boldsymbol{z}(x,y) = \sum_{i=1}^{M} \boldsymbol{\phi}_i(x,y) \ \ s.t. \ \Delta_x \boldsymbol{\phi}_i = \boldsymbol{A}\hat{\boldsymbol{\phi}}_i, \Delta_y \boldsymbol{\phi}_i = \boldsymbol{B}\hat{\boldsymbol{\phi}}_i,$$
$$(3)$$

where all FINOLA solutions $\{\boldsymbol{\phi}_i\}$ share the matrices $\boldsymbol{A}$ and $\boldsymbol{B}$. The resulting feature map $\boldsymbol{z}$ satisfies $\Delta_x \boldsymbol{z} = \boldsymbol{Q}\Delta_y \boldsymbol{z}$ (Eq. 2), but it no longer adheres to the FINOLA constraint ($\Delta_x \boldsymbol{z} \neq \boldsymbol{A}\hat{\boldsymbol{z}}, \Delta_y \boldsymbol{z} \neq \boldsymbol{B}\hat{\boldsymbol{z}}$). Notably, the vanilla FINOLA corresponds to a special case $M = 1$.

This approach can be implemented by expanding FINOLA from a single path to multiple paths. As illustrated in Figure 3, an image undergoes encoding into $M$ vectors $\{\boldsymbol{q}_1...\boldsymbol{q}_M\}$, with each vector subjected to the FINOLA process. Each path corresponds to a special solution $\boldsymbol{\phi}_i$ in Eq. 3. Subsequently, the resulting feature maps are aggregated to reconstruct the original image. Importantly, all these paths share the same set of parameters. Our experiments have validated the effectiveness of this approach, showing that the reconstruction PSNR improves as the number of paths increases.

**One-way wave equations after diagonalization:** Empirically, we consistently observed that the learned matrix $\boldsymbol{Q}$ is *diagonalizable* ($\boldsymbol{Q} = \boldsymbol{V}\boldsymbol{\Lambda}\boldsymbol{V}^{-1}$) across various training configurations. As a result, channels become decorrelated when projecting the feature map $\boldsymbol{z}$ by the inverse of eigenvectors: $\boldsymbol{\zeta}(x,y) = \boldsymbol{V}^{-1}\boldsymbol{z}(x,y)$, which modifies Eq. 2 to:

$$\Delta_x \boldsymbol{\zeta} = \boldsymbol{\Lambda}\Delta_y \boldsymbol{\zeta}, \ where \ \boldsymbol{\Lambda} = \mathrm{diag}(\lambda_1, \lambda_2, \ldots, \lambda_C). \quad (4)$$

where channels in $\boldsymbol{\zeta}$ are decorrelated. Each channel $\zeta_k$ follows an independent linear partial difference equation $\Delta_x \zeta_k = \lambda_k \Delta_y \zeta_k$. It is a finite approximation of a one-way wave equation (or transportation equation) as follows:

$$\frac{\partial \zeta_k}{\partial x} = \lambda_k \frac{\partial \zeta_k}{\partial y}, \quad (5)$$

where $\lambda_k$ is the $k^{th}$ eigenvalue in $\boldsymbol{\Lambda}$. For each channel $\zeta_k$, the rate of change along the $x$-axis is $\lambda_k$ times the rate of change along the $y$-axis. Its solution takes the form $\mathcal{F}_k(\lambda_k x + y)$, where $\mathcal{F}_k(\cdot)$ can be any differentiable function. Typically, a one-way wave equation involves time $t$ as $\frac{\partial u}{\partial x} = c\frac{\partial u}{\partial t}$; here, we replace $t$ with $y$.

**Key insight:** The amalgamation of Eqs. 1, 3, and 5 empirically reveals an insight into understanding images: images *may share a set of one-way wave equations* in the latent feature space. Each image corresponds to *a distinct solution* that can be generated from its *associated initial condition*, as illustrated in Figure 1.

Table 1: **Reconstruction PSNR across various resolutions.** Performance drops slightly at higher resolutions which have significant fewer parameters in the following upsampling and convolution layers.

| Resolution | upsample/conv #Params | Single-path $1\times3072$ | Multi-path $4\times1024$ |
|---|---|---|---|
| $8\times8$ | 25.3M | 25.4 | 25.9 |
| $16\times16$ | 18.5M | 25.8 | 26.2 |
| $32\times32$ | 9.6M | 25.8 | 26.2 |
| $64\times64$ | 7.9M | 25.7 | 26.1 |
| $128\times128$ | 1.7M | 25.3 | 25.4 |
| $256\times256$ | 1.2M | 24.6 | 24.8 |

Table 2: **Comparison with simpler autoregressive baselines.** PSNR values for image reconstruction on the ImageNet-1K validation set are reported. Image size is $256\times256$. Single-path FINOLA with $C = 3072$ channels is used. $\ddagger$ denotes the use of position embedding.

| Autoregression | Resolution | |
|---|---|---|
| | $16\times16$ | $64\times64$ |
| Repetition | 16.1 | 13.3 |
| Repetition$^{\ddagger}$ | 20.2 | 21.2 |
| Linear | 25.4 | *not converge* |
| **Norm+Linear** | **25.8** | **25.7** |

Table 3: **Comparison with norm+nonlinear.** PSNR values for image reconstruction are reported. The norm+nonlinear baseline replaces the *linear* model in FINOLA with two MLP layers incorporating GELU activation in between.

| Autoregression | $C$=512 | $C$=1024 | $C$=3072 |
|---|---|---|---|
| Norm+Nonlinear | **22.4** | **23.8** | **25.8** |
| **Norm+Linear** | 22.2 | 23.7 | **25.8** |

Table 4: **Comparison between normalization models.** PSNR values for image reconstruction are reported. Layer-norm is significantly better than batch-norm on the validation set.

| Normalization | Training | Validation |
|---|---|---|
| Batch-Norm | 25.1 | 16.3 |
| Layer-Norm | **25.5** | **25.8** |

Both FINOLA solution and initial condition can be easily transformed to the new feature space $\zeta$. The transformed initial condition is $z(\frac{W}{2}, \frac{H}{2}) = q$. The FINOLA solution in Eq. 3 is transformed as follows:

$$\zeta(x,y) = \sum_{i=1}^{M} \psi_i(x,y), \ \Delta_x\psi_i = H_A\hat{\psi}_i, \ \Delta_y\psi_i = H_B\hat{\psi}_i, \tag{6}$$

where the transformed FINOLA series $\psi_i$ and matrices $H_A$ and $H_B$ are computed by multiplying the inverse of eigenvectors $V^{-1}$ before $\phi_i$, $A$, and $B$, respectively:

$$\psi_i = V^{-1}\phi_i, \ \ H_A = V^{-1}A, \ \ H_B = V^{-1}B,$$
$$\hat{\psi}_i = \frac{(CI - J)V\psi_i}{\sqrt{\psi_i^T V^T (CI - J)V\psi_i}}, \tag{7}$$

where $C$ represents the number of channels $\psi_i(x,y) \in \mathbb{C}^C$, $I$ and $J$ are the identity and all-ones matrices respectively. Unlike the normalization of $\hat{\phi}_i$ in Eq. 3, which simply divides the standard deviation after subtracting the mean, the derivation of normalization $\hat{\psi}_i$ is shown in Appendix E.1.

**Implementation clarification:** We clarify that the interpretation via one-way wave equation does *not* guide training, but reveals an insight through post-training processing. Specifically, wave speeds, denoted as $\Lambda$, are *not explicitly* learned during training. Instead, they are computed post-training by diagonalizing trainable matrices $A$ and $B$ as

$AB^{-1} = V\Lambda V^{-1}$. Examination of the eigenvalues in $\Lambda$ and eigenvectors in $V$ across various trained models confirms their complex nature ($\Lambda, V \in \mathbb{C}^{C\times C}$). Please refer to Appendix B.9 for enforcing real-valued wave speed.

Additionally, it's noteworthy that the diagonalizability of $AB^{-1}$ is *not* guaranteed since matrices $A$ and $B$ are learned from training loss without imposed constraints. However, in practice, our experiments indicate that non-diagonalizable matrices rarely occur. This observation suggests that the set of matrices resistant to diagonalization is sufficiently small through the learning process.

## 4. Experiments on Image Reconstruction

We evaluate our FINOLA (single and multiple paths) for image reconstruction on ImageNet-1K (Deng et al., 2009). The default image size is $256\times256$. Our models are trained on the training set and subsequently evaluated on the validation set. Please refer to Appendix B.2 for model and training details, and Appendix B.6–B.10 for additional ablations, experimental results and visualization.

### 4.1. Main Properties

**FINOLA across various resolutions:** Table 1 shows consistent PSNR scores across various feature map resolutions for both single-path and multi-path FINOLA. Minor performance reduction occurs at $128\times128$ and $256\times256$ due to

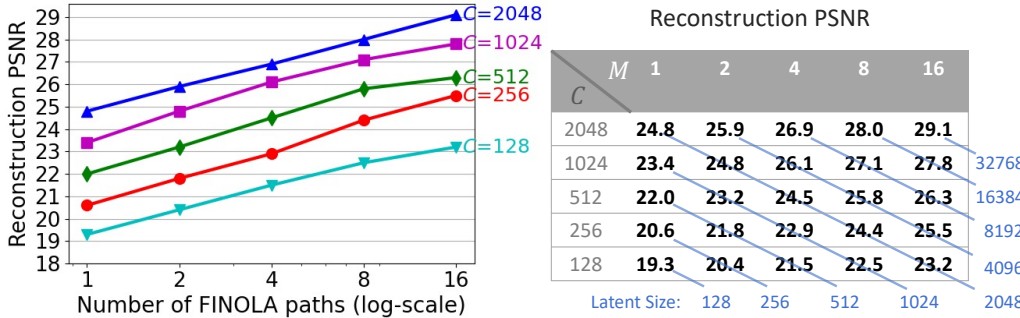

Figure 5: **Reconstruction PSNR for multi-path FINOLA.** The generated feature map has a resolution of $64 \times 64$, and the image size is $256 \times 256$. Increasing the number of paths $M$, as defined in Eq. 3, consistently enhances reconstruction PSNR across various dimensions ($C = 128$ to $C = 2048$). The blue lines in the right table represent contour lines of the latent size (equal to $MC$). PSNR remains consistent along each latent size line. Best viewed in color.

smaller decoders (1.7M and 1.2M parameters, respectively). Notably, at resolution $256 \times 256$, FINOLA is followed by only three 3x3 convolutional layers, covers a 7-pixel field of view (see Table 13 in Appendix B.2).

**Norm+Linear:** Table 2 underscores the irreplaceability of norm+linear, as simpler alternatives like repetition exhibit significantly lower PSNR, and a linear model without normalization fails to converge at higher resolutions ($64 \times 64$). Additionally, Table 3 demonstrates that replacing the linear component with a more complex 2-layer MLP yields negligible gain. Moreover, Table 4 emphasizes the important role of layer normalization, with a substantial drop in validation observed for batch normalization. These findings collectively establish that norm+linear is necessary and sufficient.

**Multi-path FINOLA:** Figure 5 shows the PSNR values for multi-path FINOLA. Increasing the number of paths $M$ consistently improves PSNR. Visual comparisons in Figure 16 at Appendix B.8 emphasize the notably enhanced image quality from single-path to multi-path FINOLA, showcasing its ability to find superior solutions within the wave equation solution space. However, multi-path also increases the latent size of initial conditions ($\sum |\boldsymbol{q}_i| = MC$). The right side of Figure 5 demonstrates a consistent PSNR along the same latent size line. This suggests that reconstruction quality is influenced not solely by the number of wave equations $C$ or the number of FINOLA paths $M$ but by their product $MC$ (the latent size). This finding enables parameter efficiency in matrices $\boldsymbol{A}$ and $\boldsymbol{B}$ by decreasing the number of channels and increasing the number of paths, which is important for large latent size. For instance, at a latent size of 16,384, single path requires 268 million parameters in matrices $\boldsymbol{A}$ and $\boldsymbol{B}$, whereas aggregating 16 FINOLA paths incurs only 1 million parameters.

**Image distribution in $q$ space:** We made three intriguing

Table 5: **Comparison with discrete cosine transform (DCT).** PSNR values for image reconstruction are reported on the ImageNet-1K validation set. $(2048 \times 16)$ indicates $C = 2048$ channels and $M = 16$ FINOLA paths. $^\dagger$ denotes using multiple initial conditions $\boldsymbol{q}_i$ at different positions instead of overlapping at the center (see Appendix B.10).

| Method | Latent ↓ | PSNR ↑ |
|---|---|---|
| DCT (top-left 1) | 3072 | 20.6 |
| **FINOLA** (multi-path) | **2048** $_{(1024 \times 2)}$ | **24.8** |
| DCT (top-left 3) | 9216 | 23.5 |
| **FINOLA** (multi-path) | **8192** $_{(1024 \times 8)}$ | **27.1** |
| DCT (top-left 6) | 18432 | 25.6 |
| **FINOLA** (multi-path) | **16384** $_{(2048 \times 8)}$ | **28.0** |
| **FINOLA** (multi-path)$^\dagger$ | **16384** $_{(2048 \times 8)}$ | **28.9** |
| DCT (top-left 10) | **30720** | 27.5 |
| **FINOLA** (multi-path) | 32768 $_{(2048 \times 16)}$ | **29.1** |
| **FINOLA** (multi-path)$^\dagger$ | 32768 $_{(2048 \times 16)}$ | **30.0** |

observations about how images are distributed in the space of the compressed vector $\boldsymbol{q}$: (a) the reconstruction from the averaged $\bar{\boldsymbol{q}}$ over 50k validation images results in a gray image (Figure 12 in Appendix B.7), (b) the space is predominantly occupied by noisy images (Figure 11 in Appendix B.7), and (c) the reconstruction from an interpolation between two embeddings, $\alpha\boldsymbol{q}_1 + (1 - \alpha)\boldsymbol{q}_2$, yields a mix-up of corresponding images (Figure 13 in Appendix B.7).

### 4.2. Comparison with Previous Techniques

We compare multi-path FINOLA with widely recognized encoding/decoding methods, such as discrete cosine transform, discrete wavelet transform, auto-encoders. The comparison is based on a *similar number of latent coefficients*.

Table 6: **Comparison with discrete wavelet transform (DWT).** PSNR values for image reconstruction are reported on the ImageNet-1K validation set. $(2048 \times 16)$ indicates $C = 2048$ channels and $M = 16$ FINOLA paths. [†] denotes using multiple initial conditions at different positions instead of overlapping at the center (see Appendix B.10).

| Method | Latent ↓ | PSNR ↑ |
|---|---|---|
| DWT (scale-3 LL subband) | 3888 | 21.5 |
| DTCWT (scale-3 LL subband) | 12288 | 22.3 |
| **FINOLA** (multi-path) | **2048** $_{(1024 \times 2)}$ | **24.8** |
| DWT (scale-3 all subbands) | 15552 | 24.3 |
| DTCWT (scale-3 all subbands) | 49152 | 25.6 |
| **FINOLA** (multi-path) | **8192** $_{(1024 \times 8)}$ | **27.1** |
| DWT (scale-2 all subbands) | 55953 | 28.7 |
| DTCWT (scale-2 all subbands) | 196608 | **30.8** |
| **FINOLA** (multi-path) | **32768** $_{(2048 \times 16)}$ | 29.1 |
| **FINOLA** (multi-path)[†] | **32768** $_{(2048 \times 16)}$ | 30.0 |

Table 7: **Comparison with convolutional auto-encoder (Conv-AE).** FINOLA (multi-path) achieves a higher PSNR compared to Conv-AE with the same latent size, while using significantly fewer parameters in the decoder. Both methods employ the same Mobile-Former encoder, and the same upsampling/convolution layers after the feature map $z$ is generated at resolution $64 \times 64$.

| Method | Latent | Param↓ | PSNR↑ |
|---|---|---|---|
| Conv-AE | 2048 | 35.9M | 24.6 |
| **FINOLA** | 2048 $_{(1024 \times 2)}$ | **16.6M** | **24.8** |
| Conv-AE | 8192 | 61.9M | 26.0 |
| **FINOLA** | 8192$_{(1024 \times 8)}$ | **16.6M** | **27.1** |

**Comparison with discrete cosine transform (DCT) (Ahmed et al., 1974):** Table 5 compares FINOLA with DCT. DCT is conducted per $8 \times 8$ image block, and the top-left $K$ coefficients (in zig-zag manner) are kept, while the rest are set to zero. We choose four $K$ values (1, 3, 6, 10) for comparison. Clearly, multi-path FINOLA achieves a higher PSNR with a similar latent size.

**Comparison with discrete wavelet transform (DWT/DTCWT) (Strang, 1989; Daubechies, 1992; Vetterli & Kovacevic, 2013):** We compare FINOLA with DWT and DTCWT in Table 6. Three scales are chosen for wavelet decomposition. The comparisons are organized into three groups: (a) using only the LL subband at the coarsest scale (scale 3), (b) using all subbands (LL, LH, HL, HH) at the coarsest level, and (c) using all subbands at the finer scale (scale 2). Our method outperforms DWT

Table 8: **Comparison with JPEG on end-to-end compression.** A single-path FINOLA model with $C = 3072$ channels is compared to JPEG compression end-to-end on ImageNet (Deng et al., 2009) and Kodak (Company, 1999) datasets. FINOLA has a much cheaper pipeline, i.e. uniform quantization per channel *without* additional coding of the quantized bits, but achieves superior performance compared to JPEG.

| Method | ImageNet | | Kodak | |
|---|---|---|---|---|
| | Bit/Pixel↓ | PSNR↑ | Bit/Pixel↓ | PSNR↑ |
| JPEG | 0.50 | 24.5 | 0.20 | 24.0 |
| **FINOLA** | **0.19** | **24.9** | **0.19** | **25.6** |

and DTCWT in terms of PSNR for the first two groups, achieving at a smaller latent size. In the last group, while FINOLA's PSNR is lower than DTCWT, its latent size is significantly smaller (more than 6 times smaller).

**Comparison with convolutional auto-encoder (Masci et al., 2011; Ronneberger et al., 2015; Rombach et al., 2021):** Table 7 presents a comparison between our method and convolutional autoencoder (Conv-AE) concerning image reconstruction, measured by PSNR. Both approaches share the same Mobile-Former (Chen et al., 2022) encoder and have identical latent sizes (2048 or 8192). In our method, multi-path FINOLA is initially employed to generate a $64 \times 64$ feature map, followed by upsampling+convolution to reconstruct an image with size $256 \times 256$. On the other hand, Conv-AE employs a deeper decoder that utilizes upsampling+convolution from the latent vector to reconstruct an image. Please see Table 15 in Appendix B.3 for details in architecture comparison. Our method has significantly fewer parameters in the decoder. The results highlight the superior performance of our method over Conv-AE, indicating that a single-layer *FINOLA* is more effective than a multi-layer upsampling+convolution approach. The comparison with auto-encoding (first stage) in generative models (e.g. Stable Diffusion (Rombach et al., 2021)) is shown in Table 16 in Appendix B.4.

### 4.3. Comparison with JPEG on Image Compression

In Table 8, we compare FINOLA (single path with 3072 channels) with JPEG for image compression. Remarkably, by employing only uniform quantization per channel without further coding of the quantized bits, FINOLA achieves higher PSNR values with lower bits per pixel on both the ImageNet and Kodak (Company, 1999) datasets.

### 4.4. Comparison with AE in Generative Models

Table 9 presents a comparison of FINOLA with the first stage (learning an autoencoder and vector quantization in

Table 9: **Comparisons with the first stage of multiple generative methods** (learning an autoencoder and vector quantization in the latent space), assessed using both PSNR and Fréchet Inception Distance (FID) metrics.

| Method | Latent Size | #Channels | Loss | | | | FID ↓ | PSNR ↑ |
|---|---|---|---|---|---|---|---|---|
| | | | Logit-Laplace | $L_2$ | Perceptual | GAN | | |
| DALL-E (Ramesh et al., 2021) | $16 \times 16$ | – | ✓ | | | | 32.00 | 22.8 |
| VQGAN (Esser et al., 2021) | $16 \times 16$ | 256 | | | ✓ | ✓ | 4.98 | 19.9 |
| ViT-VQGAN (Yu et al., 2022a) | $32 \times 32$ | 32 | ✓ | ✓ | ✓ | ✓ | 1.28 | – |
| Stable Diffusion (Rombach et al., 2021) | $16 \times 16$ | 16 | | | ✓ | ✓ | **0.87** | 24.1 |
| **FINOLA** | $1 \times 1$ | 3072 | | ✓ | | | 27.82 | **25.8** |

the latent space) of multiple generative methods, assessed using both PSNR and Fréchet Inception Distance (FID) metrics. While FINOLA achieves good performance in terms of PSNR, its results are less competitive with respect to FID. This divergence is a consequence of our deliberate choice to employ the $L_2$ loss function. We intentionally prioritized a straightforward loss function to emphasize that FINOLA's efficacy does not depend on complex reconstruction loss formulations. To further optimize FID scores, we acknowledge the potential benefits of incorporating perceptual loss functions and Generative Adversarial Network (GAN) losses, as demonstrated in models such as VQGAN (Esser et al., 2021), ViT-VQGAN (Yu et al., 2022a), Stable Diffusion (Rombach et al., 2021). We also leave more extensive comparisons with recent generative models (Shocher et al., 2024; Ismail et al., 2024) to future work.

It is crucial to reiterate that FINOLA's primary objective differs from that of a generative model. Our focus is not on image generation per se, but rather on introducing a novel perspective for understanding images (first-order norm+linear autoregression).

## 5. Application on Self-Supervised Learning

FINOLA can be applied to self-supervised learning through a straightforward masked prediction task, which we refer to as *Masked FINOLA* to distinguish it from the vanilla FINOLA. Please refer to Appendix C for details of masked prediction, network structure, training setup, and additional experiments. Our key findings include:

**Comparable performance:** Masked FINOLA demonstrates comparable performance to established baselines, e.g. MAE (He et al., 2021) and SimMIM (Xie et al., 2022), on ImageNet fine-tuning (see Table 10), as well as linear probing (see Table 25 in Appendix C.4), while maintaining lower computational requirements.

**Robust task-agnostic encoders:** Pre-training with Masked FINOLA, followed by fine-tuning on ImageNet-1K (IN-1K), consistently outperforms IN-1K supervised pre-training in both ImageNet classification and COCO object detection (see Figure 6). The gains in object detection are substantial,

Table 10: **Comparison with previous self-supervised methods on ImageNet-1K fine-tuning**. The baseline methods includes MoCo-v3 (Chen et al., 2021), MAE-Lite (Wang et al., 2022), UMMAE (Li et al., 2022b), MAE (He et al., 2021), and SimMIM (Xie et al., 2022). Three Mobile-Former backbones of varying widths are used, followed by a decoder with 4 transformer blocks.

| Method | Model | MAdds↓ | #Params↓ | Top-1↑ |
|---|---|---|---|---|
| MoCo-v3 | ViT-Tiny | 1.2G | **6M** | 76.8 |
| MAE-Lite | ViT-Tiny | 1.2G | **6M** | 78.0 |
| **FINOLA** | MF-W720 | **0.7G** | 7M | **78.4** |
| MoCo-v3 | ViT-S | 4.6G | 22M | 81.4 |
| UM-MAE | Swin-T | 4.5G | 29M | 82.0 |
| MAE-Lite | ViT-S | 4.6G | 22M | 82.1 |
| SimMIM | Swin-T | 4.5G | 29M | **82.2** |
| **FINOLA** | MF-W1440 | **2.6G** | **20M** | 82.2 |
| MoCo-v3 | ViT-B | 16.8G | 86M | 83.2 |
| MAE | ViT-B | 16.8G | 86M | 83.6 |
| SimMIM | ViT-B | 16.8G | 86M | 83.8 |
| SimMIM | Swin-B | 15.4G | 88M | **84.0** |
| **FINOLA** | MF-W2880 | **9.9G** | **57M** | 83.9 |

ranging from 5 to 6.4 AP. Table 26 in Appendix C.5 shows that FINOLA outperforms MoCo-v2 on various tasks such as image classification, object detection, and segmentation. Notably, the encoder is *frozen* without fine-tuning on detection and segmentation tasks. Please refer to Appendix C.5 for additional experimental results.

**FINOLA vs. Masked FINOLA:** Table 30 in Appendix D.1 compares vanilla FINOLA and two masked FINOLA variants in image reconstruction and linear probing. The introduction of masking in masked FINOLA trades restoration accuracy for improved semantic representation. Geometrically, Figure 23 in Appendix D.2 illustrates masked FINOLA introduces a substantial increase in Gaussian curvature on critical feature surfaces, suggesting enhanced curvature in the latent space for capturing semantics. Computation details of Gaussian curvature are available in Appendix D.3. Additional comparisons can be found in Appendix D.1.

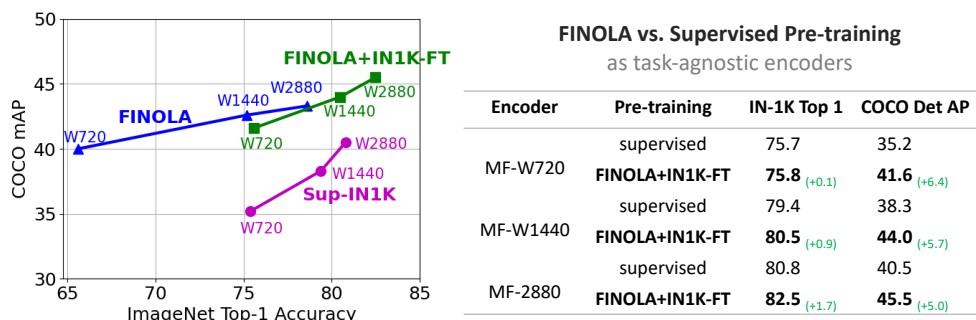

Figure 6: **Task-agnostic encoders** evaluated on ImageNet (IN-1K) classification and COCO object detection. We assess three IN-1K pretraining methods: (a) supervised (Sup-IN1K), (b) FINOLA, and (c) FINOLA with fine-tuning on IN-1K (FINOLA+IN1K-FT). The dots represent different Mobile-Former backbones. It is noteworthy that the backbone remains task-agnostic and is kept frozen during object detection. FINOLA performs lower than Sup-IN1K in classification but surpasses it in object detection. After fine-tuning on IN-1K, FINOLA+IN1K-FT shows improvements in both tasks, providing robust task-agnostic encoders.

## 6. Related Work

**Image autoregression:** Autoregression has played a pivotal role in generating high-quality images (van den Oord et al., 2016b;a; Salimans et al., 2017; Chen et al., 2018). These methods model conditional probability distributions of current pixels based on previously generated ones, evolving from pixel-level focus to latent space modeling using vector quantization (van den Oord et al., 2017; Razavi et al., 2019; Esser et al., 2021; Yu et al., 2022c). In contrast, we present a first-order norm+linear autoregression to generate feature map and reveals new insights by generalizing FINOLA as a set of one-way wave equations.

**Image transforms:** The Discrete Cosine Transform (DCT) (Ahmed et al., 1974) and Wavelet Transform (Strang, 1989; Daubechies, 1992; Vetterli & Kovacevic, 2013) are widely recognized signal processing techniques for image compression. Both DCT and wavelet transforms project images into a *complete* space consisting of *known* wave functions, in which each image has *compact* coefficients, i.e., most coefficients are close to zero. In contrast, our method offers a distinct mathematical perspective for representing images. It encodes images into a *compact* space represented by a set of *one-wave equations* with *learnable* speeds, with each image corresponding to a unique initial condition. These differences are summarized in Table 11 at Appendix B.1.

**Self-supervised learning:** Contrastive methods (Becker & Hinton, 1992; Hadsell et al., 2006; van den Oord et al., 2018; Wu et al., 2018; He et al., 2019; Chen & He, 2020; Caron et al., 2021) achieve significant progress. They are most applied to Siamese architectures (Chen et al., 2020b; He et al., 2019; Chen et al., 2020d; 2021) to contrast image similarity and dissimilarity and rely on data augmentation. (Chen & He, 2020; Grill et al., 2020) remove dissimilarity

between negative samples by handling collapse carefully. (Chen et al., 2020c; Li et al., 2021a) show pre-trained models work well for semi-supervised learning and few-shot transfer. Masked image modeling (MIM) is inspired by BERT (Devlin et al., 2019) and ViT (Dosovitskiy et al., 2021) to learn representation via masked prediction. BEiT (Bao et al., 2021) and PeCo (Dong et al., 2021) predict on tokens, MaskFeat (Wei et al., 2022) predicts on HOG, and MAE (He et al., 2021) reconstructs original pixels. Recent works explore combining MIM and contrastive learning (Zhou et al., 2022; Dong et al., 2022; Huang et al., 2022; Tao et al., 2022; Assran et al., 2022; Jiang et al., 2023) or techniques suitable for ConvNets (Gao et al., 2022; Jing et al., 2022; Fang et al., 2022). Different from using random masking in these works, FINOLA uses regular masking and simpler norm+linear prediction.

## 7. Conclusion

In this paper, we empirically revealed a form of invariance in natural images through the lens of one-way wave equations. Our findings suggest that all images may share a common set of such equations, defined by learnable propagation speeds, with each image corresponding to a unique solution determined by its initial condition. This dynamic is seamlessly implemented within an encoder-decoder framework, where the wave equations are reformulated as a first-order norm-plus-linear autoregressive (FINOLA) process. The proposed approach demonstrates strong performance in image reconstruction and shows promising potential for self-supervised learning, offering a novel perspective on the underlying structure of visual data. Looking forward, we believe that further theoretical investigations into the dynamics uncovered in this work could provide valuable insights and foundations for future research.

## Impact Statement

This paper presents work whose goal is to advance the field of Machine Learning. There are many potential societal consequences of our work, none which we feel must be specifically highlighted here.

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

Table 11: Comparison between DCT/Wavelet transform and FINOLA.

|  | DCT or Wavelet Transform | FINOLA |
|---|---|---|
| Representation | Cosine/Wavelet functions | One-way wave equations |
| Parameters | *Fixed* parameters | *Learnable* speeds |
| Encoding | Image → *coefficients* | Image → *initial conditions* |
| Compactness | Compact *coefficients* per image | Compact *space* representation |

Table 12: **Specification of Mobile-Former encoders**. "bneck-lite" denotes the lite bottleneck block (Li et al., 2021b). "M-F" denotes the Mobile-Former block and "M-F$^\downarrow$" denotes the Mobile-Former block for downsampling.

| Stage | Resolution | Block | MF-W2880 | | MF-W1440 | | MF-W720 | |
|---|---|---|---|---|---|---|---|---|
| | | | #exp | #out | #exp | #out | #exp | #out |
| token | | | 6×256 | | 6×256 | | 6×192 | |
| stem | $256^2$ | conv 3×3 | – | 64 | – | 32 | – | 16 |
| 1 | $128^2$ | bneck-lite | 128 | 64 | 64 | 32 | 32 | 16 |
| 2 | $64^2$ | M-F$^\downarrow$ | 384 | 112 | 192 | 56 | 96 | 28 |
| | | M-F | 336 | 112 | 168 | 56 | 84 | 28 |
| 3 | $32^2$ | M-F$^\downarrow$ | 672 | 192 | 336 | 96 | 168 | 48 |
| | | M-F | 576 | 192 | 288 | 96 | 144 | 48 |
| | | M-F | 576 | 192 | 288 | 96 | 144 | 48 |
| 4 | $16^2$ | M-F$^\downarrow$ | 1152 | 352 | 288 | 96 | 240 | 80 |
| | | M-F | 1408 | 352 | 704 | 176 | 320 | 88 |
| | | M-F | 1408 | 352 | 704 | 176 | 480 | 88 |
| | | M-F | 2112 | 480 | 1056 | 240 | 528 | 120 |
| | | M-F | 2880 | 480 | 1440 | 240 | 720 | 120 |
| | | M-F | 2880 | 480 | 1440 | 240 | 720 | 120 |
| | | conv 1×1 | – | 2880 | – | 1440 | – | 720 |

# A. Limitations

The major limitation of our method is that the invariance (encoded in matrices $A$ and $B$) is revealed empirically without theoretical proof. Additionally, this paper focuses on multi-path FINOLA, which represents only a subspace of the solutions to the one-way wave equations. In future work, we plan to explore the theoretical analysis of the revealed invariance and the complete solution space of the one-way wave equations.

# B. FINOLA for Image Reconstruction

In this section, we list implementation details and additional experimental results of FINOLA (single or multiple paths).

### B.1. Conceptual Comparison with DCT/Wavelet Transforms

Both DCT and wavelet transforms project images into a *complete* space consisting of *known* wave functions, in which each image has *compact* coefficients, i.e., most coefficients are close to zero. In contrast, our method offers a distinct mathematical perspective for representing images. It encodes images into a *compact* space represented by a set of *one-way wave equations* with *learnable* speeds. Each image corresponds to a unique initial condition. These differences are summarized in Table 11.

Table 13: **Upsampling and convolutional layers in FINOLA decoder**. The complexity of upsampling and convolution layers decreases as the spatial resolution of feature map (generated by FINOLA) increases from 8×8 to 256×256). "res-conv" represents a residual block (He et al., 2016) consisting of two 3x3 convolutional layers, while "up-conv" performs upsampling followed by a 3x3 convolutional layer.

| Resolution | 8×8 | | 16×16 | | 32×32 | | 64×64 | | 128×128 | | 256×256 | |
|---|---|---|---|---|---|---|---|---|---|---|---|---|
| | block | #out | block | #out | block | #out | block | #out | block | #out | block | #out |
| $8^2$ | res-conv | 512 | | | | | | | | | | |
| $16^2$ | up-conv | 512 | | | | | | | | | | |
| | res-conv | 512 | res-conv | 512 | | | | | | | | |
| $32^2$ | up-conv | 512 | up-conv | 512 | | | | | | | | |
| | res-conv | 256 | res-conv | 256 | res-conv | 256 | | | | | | |
| $64^2$ | up-conv | 256 | up-conv | 256 | up-conv | 256 | | | | | | |
| | res-conv | 256 | res-conv | 256 | res-conv | 256 | res-conv | 256 | | | | |
| $128^2$ | up-conv | 256 | up-conv | 256 | up-conv | 256 | up-conv | 256 | | | | |
| | res-conv | 128 | res-conv | 128 | res-conv | 128 | res-conv | 128 | res-conv | 128 | | |
| $256^2$ | up-conv | 128 | up-conv | 128 | up-conv | 128 | up-conv | 128 | up-conv | 128 | | |
| | res-conv | 128 | res-conv | 128 | res-conv | 128 | res-conv | 128 | res-conv | 128 | res-conv | 128 |
| | conv3×3 | 3 | conv3×3 | 3 | conv3×3 | 3 | conv3×3 | 3 | conv3×3 | 3 | conv3×3 | 3 |
| #param | 25.3M | | 18.5M | | 9.6M | | 7.9M | | 1.7M | | 1.2M | |

## B.2. Implementation Details

### B.2.1. NETWORK ARCHITECTURES

In this subsection, we provide detailed information on the network architecture components used in our study. Specifically, we describe (a) the Mobile-Former encoders, (b) the pooler to compress the feature map into a single vector, (c) the upsampling and convolutional layers employed in FINOLA decoder.

**Mobile-Former encoders:** Mobile-Former (Chen et al., 2022) is used as the encoder in our approach. It is a CNN-based network that extends MobileNet (Sandler et al., 2018) by adding 6 global tokens in parallel. To preserve spatial details, we increase the resolution of the last stage from $\frac{1}{32}$ to $\frac{1}{16}$. We evaluate three variants of Mobile-Former, which are detailed in Table 12. Each variant consists of 12 blocks and 6 global tokens, but they differ in width (720, 1440, 2880). These models serve as the encoders (or backbones) for image reconstruction, self-supervised pre-training, and evaluation in image classification and object detection tasks. For image reconstruction, we also explore two wider models, W4320 and W5760, which increase the number of channels from W2880 by 1.5 and 2 times, respectively. It's important to note that these models were manually designed without an architectural search for optimal parameters such as width or depth.

**Pooling the compressed vector $q$:** In both FINOLA and element-wise masked FINOLA, the compressed vector $q$ is obtained by performing attentional pooling (Lee et al., 2019; Yu et al., 2022b) on the feature map. This pooling operation involves a single multi-head attention layer with learnable queries, where the encoder output serves as both the keys and values.

**FINOLA decoders:** Table 13 provides the architecture details of upsampling and covolutional layers after applying FINOLA to generate feature maps $z$. The complexity of decreases as the spatial resolution increases, going from 8×8 to 256×256. FINOLA is trained for 100 epochs on ImageNet.

Table 14: **Training setting for FINOLA.**

| Config | FINOLA |
|---|---|
| optimizer | AdamW |
| base learning rate | 1.5e-4 |
| weight decay | 0.1 |
| batch size | 128 |
| learning rate schedule | cosine decay |
| warmup epochs | 10 |
| training epochs | 100 |
| image size | $256^2$ |
| augmentation | RandomResizeCrop |

### B.2.2. TRAINING SETUP

The FIOLA training settings for image reconstruction are provided in Table 14. The learning rate is scaled as $lr = base\_lr \times$ batchsize / 256.

Table 15: Architecture comparison between convolutional Auto-Encoder and FINOLA.

|  | **Auto-Encoder** | **FINOLA** |
|---|---|---|
| Encoder | same | same |
| Pooling | $2 \times 2 \times 512$ ($2 \times 2$ grid) | $2 \times 1024$ (overlap at center) |
| Upsampling to $64 \times 64 \times 1024$ | 5 conv blocks (from $2 \times 2$ to $64 \times 64$) | FINOLA |
| Upsampling to $256 \times 256 \times 13$ | same | same |
| Training setup | same | same |

### B.2.3. Training and Inference Time

**Training time:** Training the FINOLA model involves regressing dense feature maps, with computational requirements increasing with feature map size. For instance, training FINOLA to generate a $16 \times 16$ feature map with 3072 latent channels for 100 epochs on ImageNet takes approximately 8 days with 8 V100 GPUs. Extending to a larger feature map, such as $64 \times 64$, increases the training time to 18 days using the same GPU setup.

**Inference time:** In addition to training time, the runtime evaluation includes the complete inference pipeline, encompassing encoding, autoregression, and decoding, conducted on a MacBook Air with an Apple M2 CPU. We evaluated FINOLA for generating feature maps of sizes $16 \times 16$ and $64 \times 64$, with running times of 1.2 seconds and 2.6 seconds, respectively.

### B.3. Architecture Comparison with Convolutional Auto-Encoder (Convv-AE)

Table 15 presents a comparison of the architectural components between the Conv-AE and FINOLA, while their performance comparison is reported in Table 7 in Section 4.2). Both models share identical (a) encoder, (b) upsampling from resolution 64x64 to $256 \times 256$, and (c) training setup (hyper-parameters). However, they differ in their approaches to pooling and upsampling toward the resolution $64 \times 64$.

**Pooling:** Auto-encoder pools a $2 \times 2$ grid with 512 channels, while FINOLA pools two vectors with dimension 1024, both yielding the same latent size (2048). Auto-encoder pooling retains spatial information within the $2 \times 2$ grid, whereas FINOLA has no explicit spatial information as both vectors are positioned centrally for the FINOLA process.

**Upsampling to Resolution $64 \times 64$:** The auto-encoder utilizes a stack of five convolutional blocks to generate features at a resolution of $64 \times 64$. Each block consists of three $3 \times 3$ convolutional layers followed by an upsampling layer to double the resolution. In contrast, our method employs multi-path FINOLA to generate the feature map from center-placed vectors. Since FINOLA utilizes only four matrices ($A$, $B$, $A_-$, and $B_-$), it significantly reduces the number of parameters compared to the five convolutional blocks used in the auto-encoder.

**Engineering techniques:** FINOLA does not rely on any additional engineering techniques. Despite this, it slightly outperforms the auto-encoder while utilizing significantly fewer parameters. We attribute this performance to FINOLA's efficient and effective modeling of spatial transitions.

Table 16: **Comparison with auto-encoding (first stage) in generative models.** PSNR values for image reconstruction are reported on the ImageNet-1K validation set. $(2048 \times 16)$ indicates $C = 2048$ channels and $M = 16$ FINOLA paths. [†] denotes using multiple initial conditions $q_i$ at different positions instead of overlapping at the center (see Appendix B.10).

| Method | Latent ↓ | PSNR ↑ |
|---|---|---|
| DALL-E | $32 \times 32 \times -$ | 22.8 |
| VQGAN | $65536 \ _{(16 \times 16 \times 256)}$ | 19.9 |
| Stable Diffusion | **4096** $_{(16 \times 16 \times 16)}$ | 24.1 |
| **FINOLA** (multi-path) | **4096** $_{(1024 \times 4)}$ | **26.1** |
| **FINOLA** (multi-path)[†] | **4096** $_{(1024 \times 4)}$ | **26.7** |
| Stable Diffusion | $12288 \ _{(64 \times 64 \times 3)}$ | 27.5 |
| **FINOLA** (multi-path) | **8192** $_{(1024 \times 8)}$ | 27.1 |
| **FINOLA** (multi-path)[†] | **8192** $_{(1024 \times 8)}$ | **28.0** |
| Stable Diffusion | $32768 \ _{(128 \times 128 \times 2)}$ | **30.9** |
| **FINOLA** (multi-path) | $32768 \ _{(2048 \times 16)}$ | 29.1 |
| **FINOLA** (multi-path)[†] | $32768 \ _{(2048 \times 16)}$ | 30.0 |

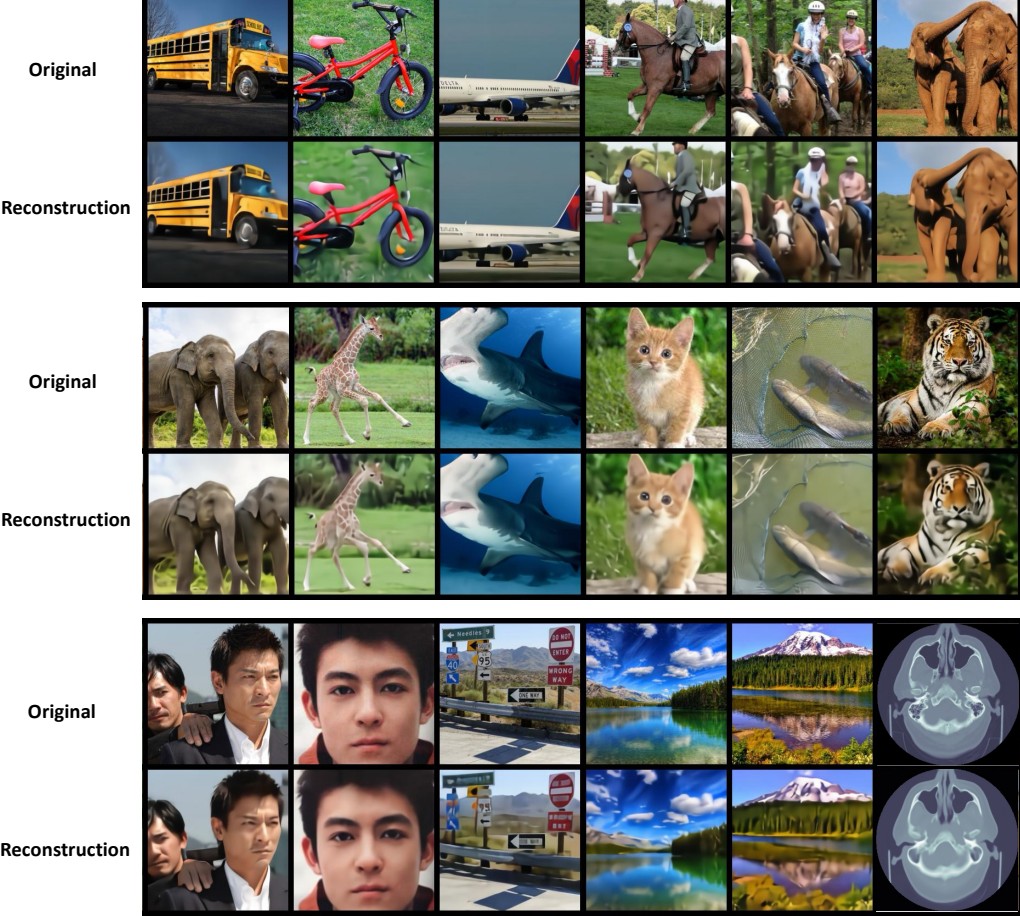

Figure 7: **Image reconstruction examples.** FINOLA works well to reconstruct diverse categories such as natural scenes, human portraits, facial images, animal photographs, air and land transportation, and medical images. Best viewed in color.

## B.4. Comparison with Auto-Encoder in Generative Models

Table 16 provides a detailed comparison of FINOLA's performance against the first stage (autoencoding) of VQGAN and Stable Diffusion in image reconstruction, evaluated on ImageNet-Val with 256x256 images. It's important to note that the first stage of VQGAN and Stable Diffusion focuses solely on auto-encoding and does not involve the generation process (e.g., diffusion process).

This comparison underscores FINOLA's performance across varying latent dimensions and its effectiveness in comparison to other methods. Although FINOLA falls behind Stable Diffusion at the largest latent dimension (32768), it operates in a more challenging setup. While FINOLA outputs a single vector after encoding, positioned at the center to generate feature maps through the FINOLA process, spatial information is not explicitly retained. In contrast, the encoder in Stable Diffusion produces a high-resolution grid (128x128) where spatial information is highly preserved.

Introducing spatial information in FINOLA by scattering the initial positions of multiple FINOLA paths (rather than overlapping at the center) enhances the reconstruction quality by 0.6-0.9 PSNR. However, due to scattering initial positions at only 16 locations, the preservation of spatial information remains constrained compared to Stable Diffusion's 128x128 grid. Consequently, while this enhancement closes the gap in performance (PSNR 30.0 vs 30.9), it still falls short of Stable Diffusion's spatial fidelity.

## B.5. Additional Reconstructed Examples

Additional FINOLA reconstructed images, encompassing diverse categories such as natural scenes, human portraits, facial images, animal photographs, air and land transportation, and medical images, are shown in Figure 7.

Table 17: **Image reconstruction ablation experiments** on ImageNet-1K. We report PSNR on the validate set. The reconstruction quality correlates to (a) the number of channels in the latent space and (b) complexity of encoder. Default settings are marked by $^\dagger$.

| #Channels | 4096 | 3072$^\dagger$ | 2048 | 1024 | 512 | 256 | 128 | 64 |
|---|---|---|---|---|---|---|---|---|
| PSNR | 25.9 | 25.8 | 25.1 | 23.7 | 22.2 | 20.8 | 19.4 | 18.2 |

*(a) Number of channels in latent space.*

| Encoder | 67.6M | 43.5M | 25.0M$^\dagger$ | 12.0M | 5.0M |
|---|---|---|---|---|---|
| PSNR | 26.1 | 26.0 | 25.8 | 25.1 | 24.4 |

*(b) Model size of encoders.*

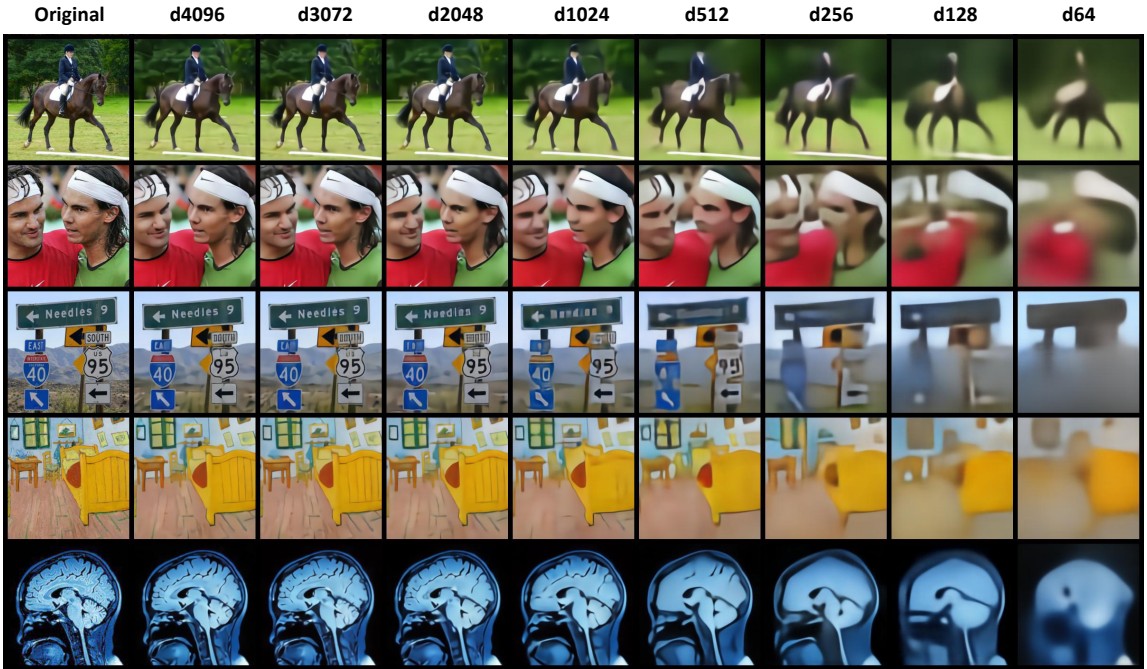

Figure 8: **Impact of the number of channels on image reconstruction quality.** The leftmost column shows the original images. The number of channels in the latent space, decreasing from 4096 to 64 from the left to right, controls the reconstruction quality. Best viewed in color.

### B.6. Ablation Studies of Single Path FINOLA

**The number of channels in the latent space is crucial.** Table 17-(a) presents the PSNR values for various latent space dimensions, while Figure 8 showcases the corresponding reconstructed examples. The image quality is noticeably poor when using only 64 channels, resulting in significant loss of details. However, as the number of channels increases, more details are successfully recovered. Using more than 3072 channels yields reasonably good image quality, achieving a PSNR of 25.8.

**The model size of encoder is less critical but also related.** As shown in Figure 9 and Table 17-(b), the larger model has better image quality. But the gap is not significant. When increasing model size by 13 times from 5.0M to 67.6M, the PSNR is slightly improved from 24.4 to 26.1. Note all encoders share similar architecture (Mobile-Former with 12 blocks), but have different widths.

**The position of $q$ is not critical:** Figure 10 showcases the reconstructed samples obtained by placing the compressed vector $q$ at different positions, including the center and four corners. The corresponding peak signal-to-noise ratio (PSNR) values on the ImageNet validation set are provided at the bottom. While placing $q$ at the center yields slightly better results compared to corner positions, the difference is negligible. It is important to note that each positioning corresponds to its own pre-trained model with non-shared parameters.

| | Original | 67.6M | 43.5M | 25.0M | 12.0M | 5.0M |
|---|---|---|---|---|---|---|

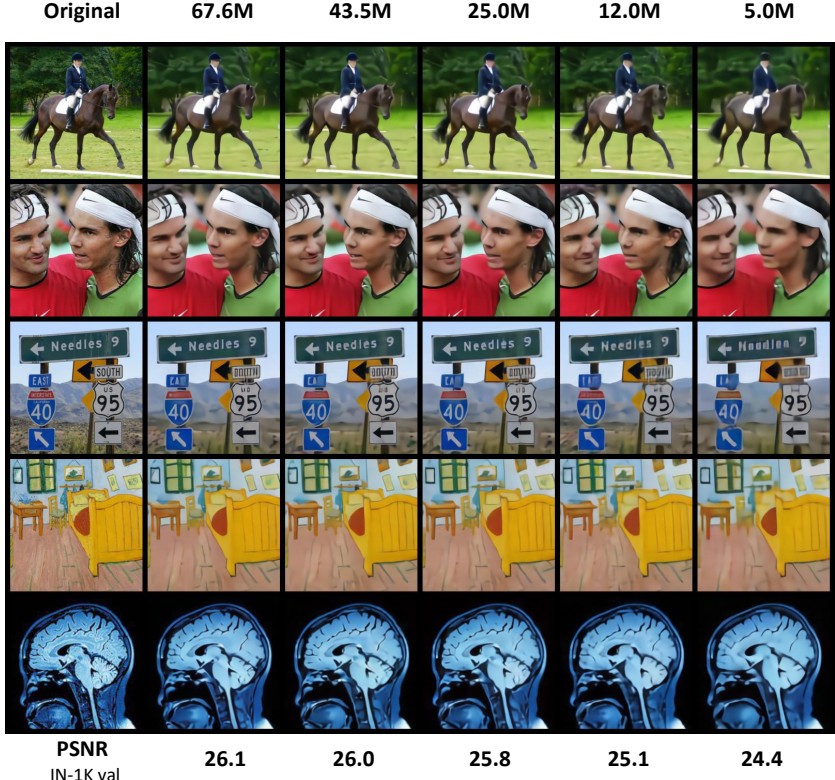

| PSNR
IN-1K val | | 26.1 | 26.0 | 25.8 | 25.1 | 24.4 |
|---|---|---|---|---|---|---|

Figure 9: **Impact of encoder size on image reconstruction quality:** The image reconstruction quality shows a slight improvement as the size of the encoder increases. Even with a small encoder containing 5 million parameters (right column), it effectively compresses an image into a single vector capable of reconstructing the entire image. Best viewed in color.

### B.7. Inspecting the Image Distribution in $q$ Space

In this subsection, we list main observations and analysis in the space of the compressed vector $q$ (named embedding space). This will help us to understand how images are distributed in the embedding space. In this subsection, we use single-path FINOLA with $C = 3072$ channels.

**Three observations:** Below we list three observations that reveal properties of the embedding space.

*Dominance of noisy images in the space:* To analyze the distribution of images in the embedding space, we collected $q$ vector for all 50,000 images from the ImageNet validation set and computed their statistics (mean and covariance). By sampling embeddings based on these statistics and reconstructing images, we consistently observed the emergence of similar noisy patterns, as depicted in Figure 11. This observation highlights the prevalence of noisy images throughout the space, with good images appearing as isolated instances surrounded by the abundance of noise.

*Averaged embedding $\bar{q}$ yields a gray image:* In Figure 12, we observe that the reconstructed image obtained from the averaged embedding $\bar{q}$, computed over 50,000 images from the ImageNet validation set, closely resembles a gray image. We further investigate the relationship between real image embeddings $q$ and the averaged embedding $\bar{q}$ through interpolations along the embedding space. As depicted in the ***left*** figure, the reconstructed images maintain their content while gradually fading into a gray image. Additionally, we extend this connection to mirror embeddings in the ***right*** figure, represented by $2q - \bar{q}$, which correspond to images with reversed colors. These findings suggest that despite the prevalence of noisy images, the line segment connecting an image embedding to the average embedding encompasses different color transformations of the same image.

*Reconstruction from interpolated embeddings:* In Figure 13, we present the reconstructed images obtained by interpolating between two image embeddings using the equation $\alpha q_1 + (1 - \alpha)q_2$. This process of embedding mixup results in a corresponding mixup of the images, allowing for a smooth transition between the two original images by varying the value

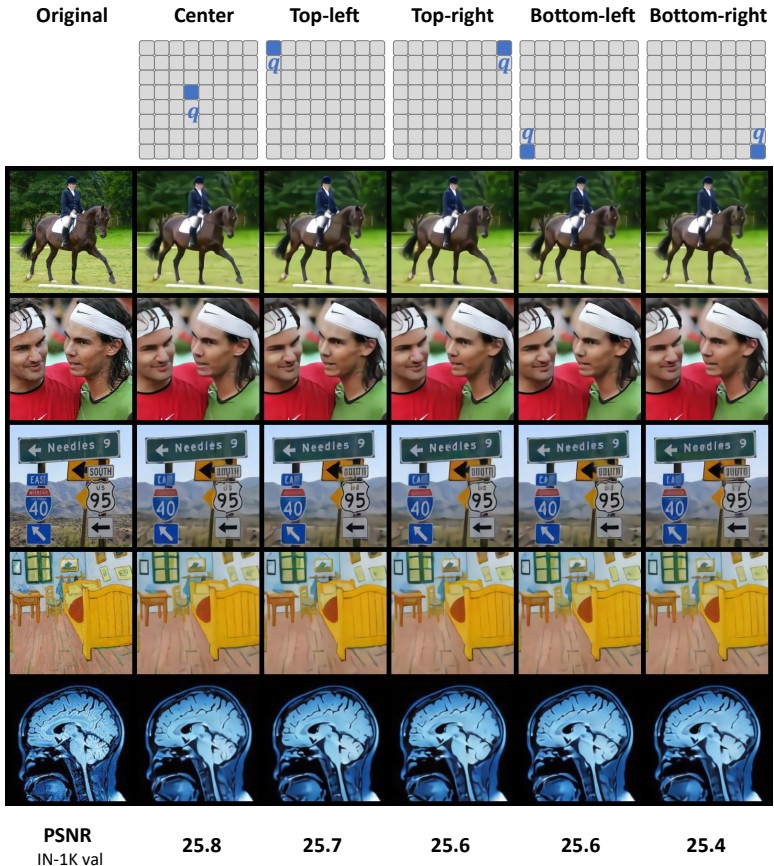

Figure 10: **Comparison of different positions of compressed vector $q$:** The quality of image reconstruction shows minimal sensitivity to the position of $q$. Placing it at the center yields slightly better results compared to corner positions. It is worth noting that each positioning has its own pre-trained model with non-shared parameters. Best viewed in color.

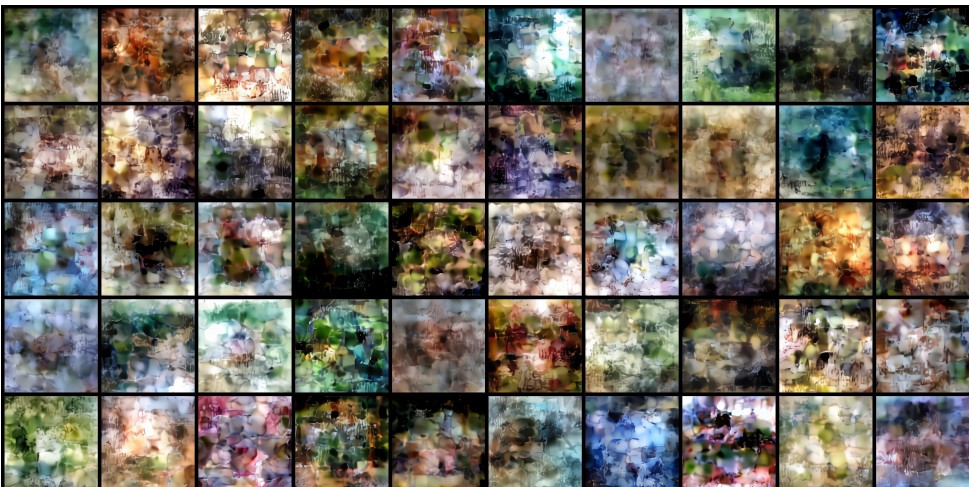

Figure 11: **Reconstruction from random samples:** The reconstructed images are generated by sampling from the statistics (mean and covariance) of compressed embeddings $q$ obtained from the ImageNet validation set, consisting of 50,000 images. Although the samples are not similar to images of Gaussian noise, they lack semantic meaning and appear as noisy images. Multiple samplings consistently yield similar noisy patterns. Best viewed in color.

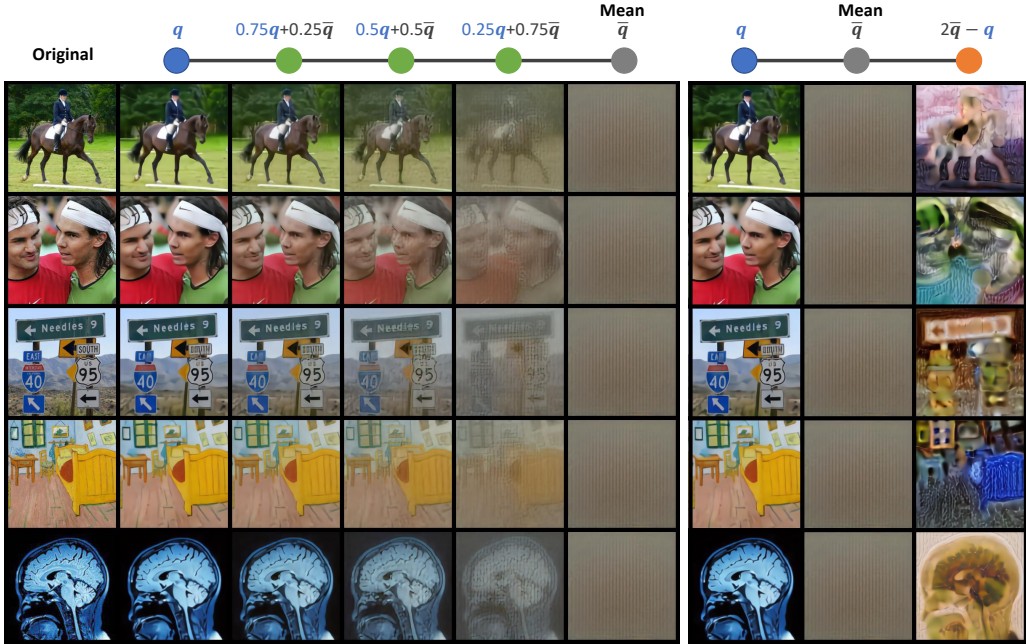

Figure 12: **Reconstruction from the average embedding $\bar{q}$:** The reconstructed image corresponding to the average embedding $\bar{q}$ computed from 50,000 ImageNet validation images closely resembles a gray image (shown in the right column of the left figure). In the *left* figure, we demonstrate the interpolation along a line connecting embeddings from different images to the average embedding. Notably, the reconstructed images progressively fade into a gray image. In the *right* figure, we extend the connection between an image embedding $q$ and the average embedding $\bar{q}$ to a mirror embedding $2q - \bar{q}$, corresponding to an image with reversed colors. This comparison provides insights into the nature of the embedding space. Best viewed in color.

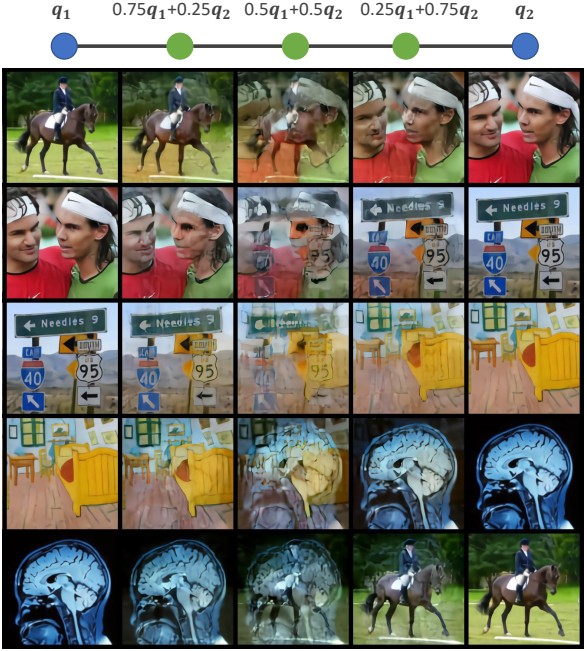

Figure 13: **Reconstruction from interpolated embeddings:** The images are reconstructed by interpolating embeddings of two images, $\alpha q_1 + (1 - \alpha)q_2$. Although the mixed embedding passes through a non-linear network that includes FINOLA and a multi-layer decoder, it leads to mixing up images as output. Best viewed in color.

**Original Images**

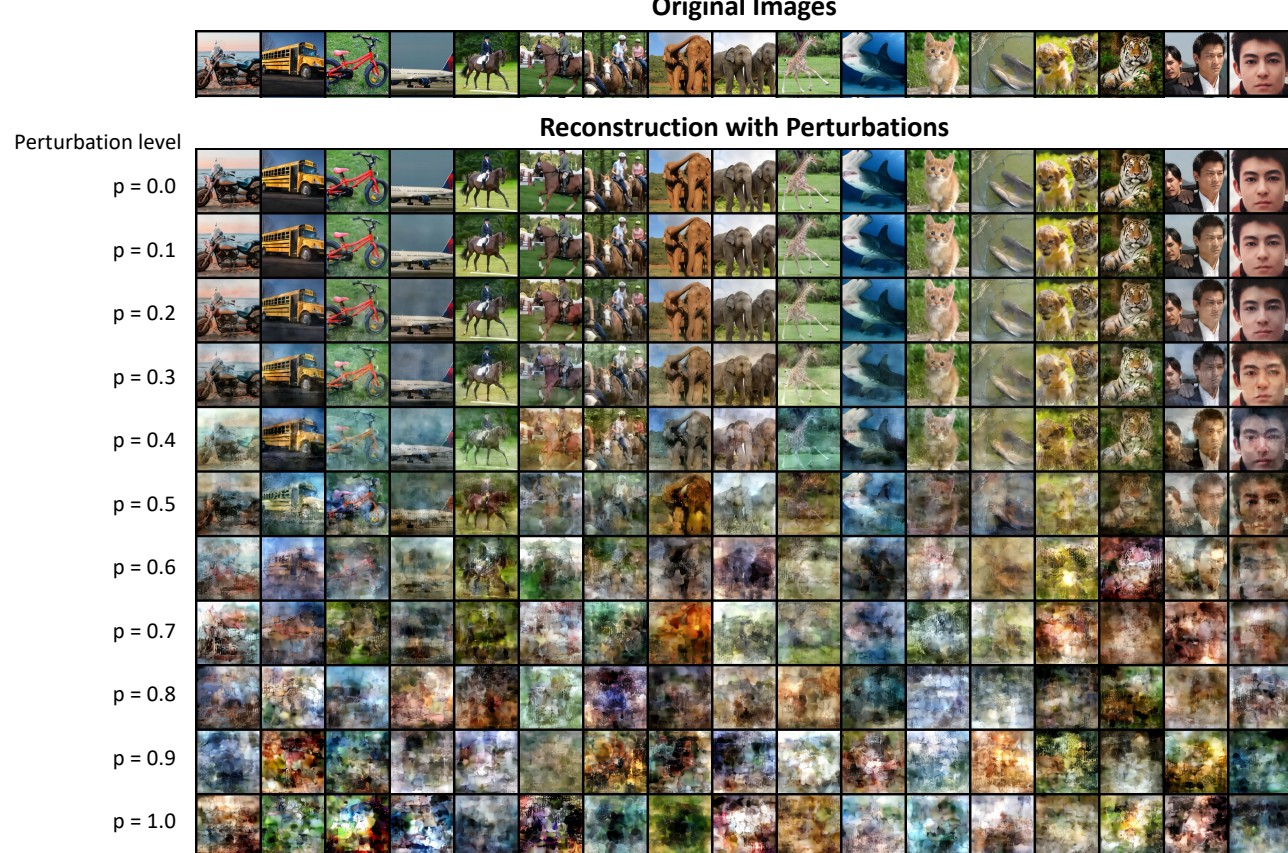

Figure 14: **Sensitivity to small perturbations:** Perturbed representations ($q_p$) were generated by linear interpolation between an image representation ($q_i$) and a randomly sampled noise vector ($q_n$): $q_p = (1 - p)q_i + pq_n$. The interpolation parameter, $p$, was varied from 0 (no perturbation) to 1.0 (full perturbation) in increments of 0.1. The perturbed vectors ($q_p$) were then processed by FINOLA and the decoder to reconstruct the images. Best viewed in color.

of $\alpha$. However, it is important to note that the resulting reconstruction may not precisely match the simple mixup of the original images, represented by $\alpha I_1 + (1 - \alpha)I_2$.

Combining the three observations discussed above, our findings suggest that the presence of noisy images in Figure 11 indicates the mixing of multiple surrounding images. As the number of image embeddings involved in the mixing process increases, the resulting reconstructions tend to resemble a gray image, as depicted in Figure 12.

**Sensitivity to small perturbations:** To assess FINOLA's sensitivity to perturbations, we conducted an experiment where perturbations were introduced into the compressed representation space. Specifically, perturbed representations ($q_p$) were generated by linear interpolation between an image representation ($q_i$) and a randomly sampled noise vector ($q_n$): $q_p = (1 - p)q_i + pq_n$. The interpolation parameter, $p$, was varied from 0 (no perturbation) to 1.0 (full perturbation) in increments of 0.1. The perturbed vectors ($q_p$) were then processed by FINOLA and the decoder to reconstruct the images. The result images are shown in Figure 14, demonstrate that FINOLA exhibits robustness to small perturbations ($p = 0.1$ or $p = 0.2$). However, the quality of the reconstruction degrades as the perturbation level increases.

**Principle component analysis (PCA):** The reconstruction results shown in Figure 15 are obtained using PCA with the top-$K$ principle components. These components correspond to the largest $K$ eigenvalues of the covariance matrix computed from 50,000 image embeddings in the ImageNet validation set. The principle components capture essential information, starting with color and layout, and gradually encoding finer image details as more components are included in the reconstruction process.

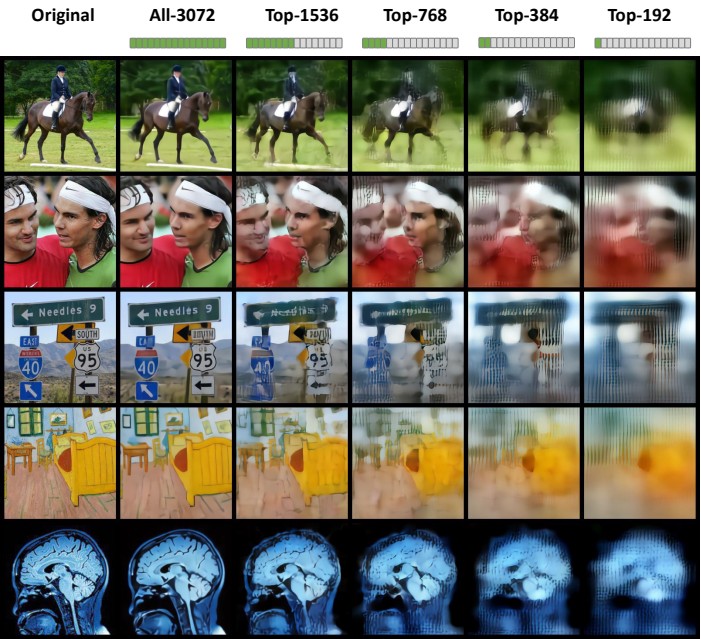

Figure 15: **Reconstruction from top principle components:** The top-$K$ principle components correspond to the largest $K$ eigenvalues of the covariance matrix computed from 50,000 image embeddings in the ImageNet validation set. With a selection of top-192 components (the right column), the color and layout of the images are primarily determined, but the resulting reconstructions appear blurred with noticeable loss of details. As more principle components are incorporated, the finer details are gradually restored. Best viewed in color.

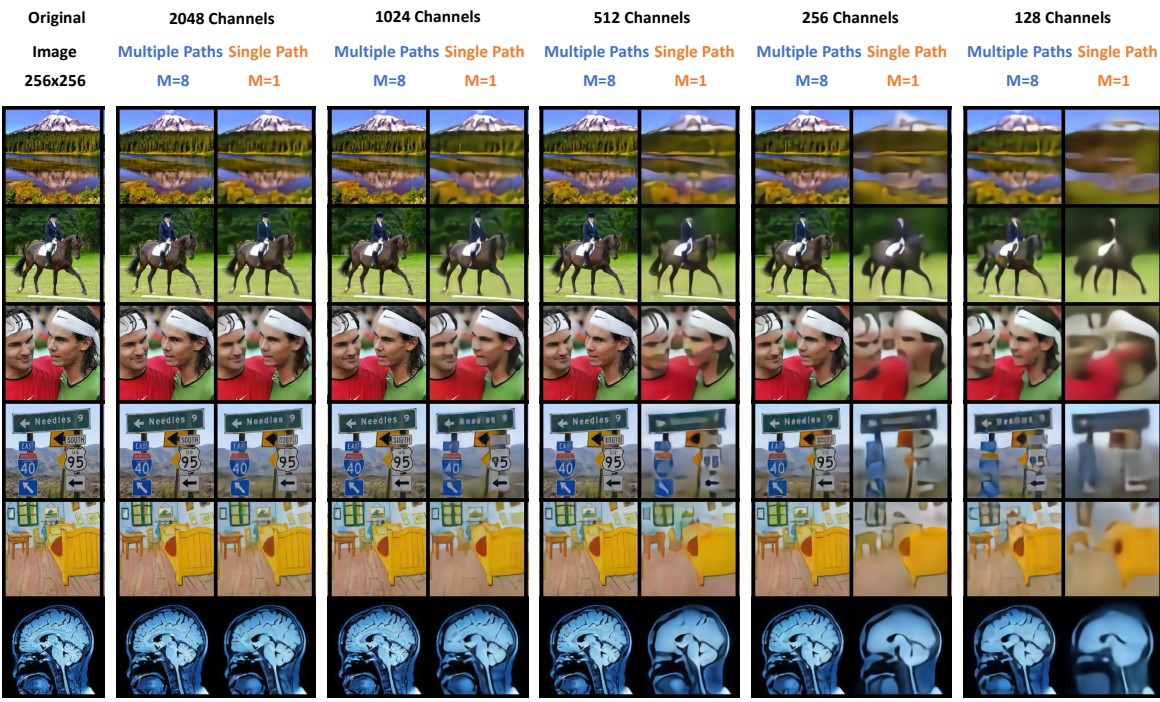

Figure 16: **Multiple paths vs. Single path:** Summing $M = 8$ FINOLA solutions $\phi_i$ (as in Eq. 3) yields superior image reconstruction quality compared to the single path counterpart. This trend holds across various dimensions (from $C = 128$ to $C = 2048$). Resolution of feature map $z$ is set to 64×64, with an image size of 256×256. Best viewed in color.

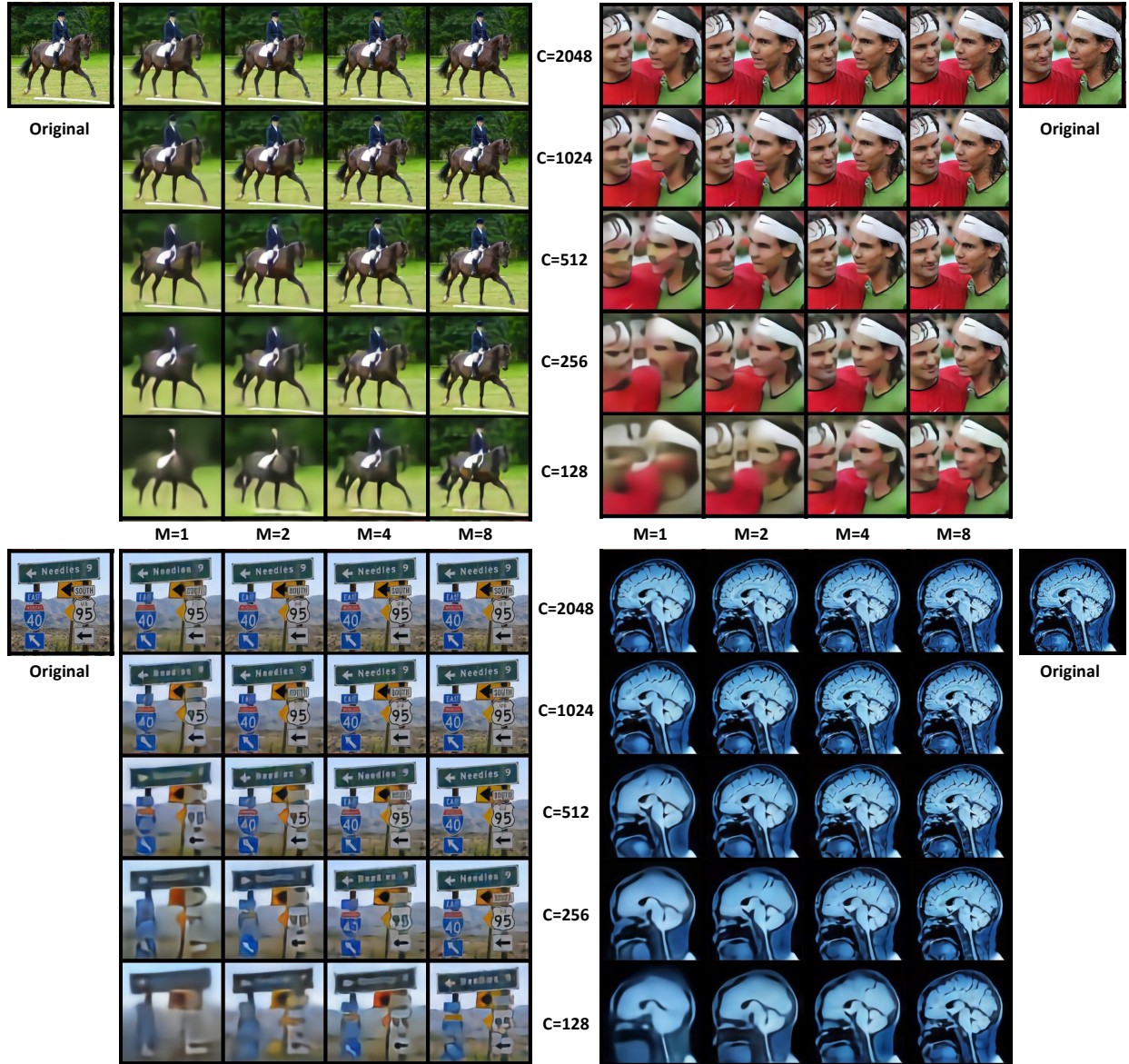

Figure 17: **Reconstruction examples for varying numbers of channels ($C$) and FINOLA paths ($M$):** Increasing the number of paths, as per Eq. 3, consistently enhances image quality across different dimensions ($C = 128$ to $C = 2048$), affirming the relaxation of FINOLA constraints. Feature resolution ($z$) is 64×64, and image size is 256×256. Best viewed in color.

## B.8. Visual Comparison between Single-path and Multi-path FINOLA

**Single vs. Multiple paths:** Figure 16 visually demonstrates that multiple paths $M = 8$ exhibit markedly superior image quality compared to the single path counterpart ($M = 1$).

**Reconstruction examples for varying number of channels $C$ and paths $M$:** Figure 17 illustrates the reconstruction examples obtained for different combinations of channel counts (or number of one-way wave equations $C = 128, 256, 512, 1024, 2048$) and the number of FINOAL paths ($M = 1, 2, 4, 8$). These results correspond to the experiments in Figure 5, as discussed in Section 4.1.

Notably, a consistent trend emerges where increasing the value of $M$ consistently enhances image quality. This trend remains consistent across various equation counts, ranging from $C = 128$ to $2048$. This observation underscores the efficacy of relaxing the FINOLA constraint by FINOLA series, as detailed in Section 3.

Table 18: **Inspection of real-valued wave speeds:** (a) PSNR values for image reconstruction with varying wave speeds (complex, real, all-one) on the ImageNet-1K validation set, with the symbol ‡ denoting the use of position embedding. The number of wave equations (or feature map dimension) is set $C = 1024$, and the number of FINOLA paths is set $M = 4$. (b) A comparison between all-one speed waves and feature map generation through repetition with position embedding to ensure position embedding isn't the sole dominant factor.

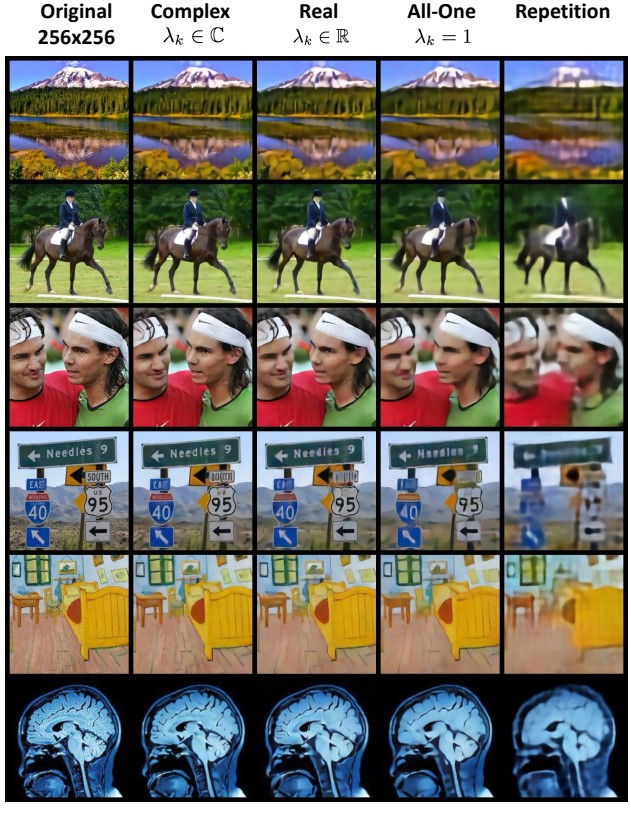

| Wave Speed | Dimension | PSNR |
|---|---|---|
| Complex $\lambda_k \in \mathbb{C}$ | 1024×4 | **26.1** |
| Real $\lambda_k \in \mathbb{R}$ | 1024×4 | 25.1 |
| All-one $\lambda_k = 1^{\ddagger}$ | 1024×4 | 23.9 |

*(a) Special cases: real and all-one speeds.*

| Feature Map Gen | Dimension | PSNR |
|---|---|---|
| Repetition | 4096 | 21.6 |
| All-one waves | 1024×4 | **23.9** |

*(b) Using position embedding.*

Figure 18: Reconstructed examples for varying wave speeds (complex, real, all-one).

## B.9. Real-valued Wave Speeds

It is worth noting that the speeds of the wave equations are generally complex numbers $\lambda_k \in \mathbb{C}$, which is also validated in the experiments. This arises because we do not impose constraints on the coefficient matrices ($\boldsymbol{A}$, $\boldsymbol{B}$) in Eq. 2. Consequently, during the diagonalization process, $\boldsymbol{A}\boldsymbol{B}^{-1} = \boldsymbol{V}\boldsymbol{\Lambda}\boldsymbol{V}^{-1}$, it is highly likely that the eigenvalues and eigenvectors will be complex numbers.

Here, we introduce two interesting cases by constraining the speeds of the one-way wave equations as follows: (a) as real numbers $\lambda_k \in \mathbb{R}$, and (b) as all equal to one $\lambda_1 = \cdots = \lambda_C = 1$.

**Real speed $\lambda_k \in \mathbb{R}$:** This is achieved by constraining matrices $\boldsymbol{H}_A$ and $\boldsymbol{H}_B$ in Eq. 6 as real diagonal matrices:

$$\boldsymbol{H}_A = \text{diag}(\alpha_1, \alpha_2, \ldots, \alpha_C), \quad \boldsymbol{H}_B = \text{diag}(\beta_1, \beta_2, \ldots, \beta_C), \quad \boldsymbol{A} = \boldsymbol{P}\boldsymbol{H}_A, \quad \boldsymbol{B} = \boldsymbol{P}\boldsymbol{H}_B. \tag{8}$$

Here, the coefficient matrices $\boldsymbol{A}$ and $\boldsymbol{B}$ in FINOLA are implemented by multiplying a real projection matrix $\boldsymbol{P}$ with diagonal matrices $\boldsymbol{H}_A$ and $\boldsymbol{H}_B$, respectively. Consequently, the speeds of the wave equations are real numbers, denoted as $\lambda_k = \alpha_k/\beta_k$.

**All-one speed $\lambda_1 = \cdots = \lambda_C = 1$:** By further constraining $\boldsymbol{H}_A$ and $\boldsymbol{H}_B$ as identity matrices, all wave equations have identical speed $\lambda_k = 1$.

$$\boldsymbol{H}_A = \boldsymbol{H}_B = \boldsymbol{I}, \quad \boldsymbol{A} = \boldsymbol{B} = \boldsymbol{P}, \quad \lambda_1 = \lambda_2 = \cdots = \lambda_C = 1. \tag{9}$$

Here, the coefficient matrices $\boldsymbol{A}$ and $\boldsymbol{B}$ in FINOLA are also identical and denoted as $\boldsymbol{P}$.

**Experimental results for real-valued wave speeds:** Table 18-(a) provides the results for real-valued and all-one wave speed, while Figure 18 displays corresponding reconstruction examples. In comparison to the default scenario using

complex-valued wave speeds, enforcing wave speeds as real numbers or setting them uniformly to one shows a slight decline in performance. Nonetheless, both real-valued speed cases still deliver reasonably good PSNR scores. Notably, the all-one wave speed configuration achieves a PSNR of 23.9. This specific configuration shares the coefficient matrix for autoregression across all four directions (up, down, left, right), creating symmetry in the feature map. To account for this symmetry, we introduced position embedding before entering the decoder.

In an effort to determine whether position embedding is the dominant factor for all-one wave speed, we conducted experiments by generating feature maps using both repetition and position embedding, with the same dimention (4096). This approach falls short of the all-one wave speed configuration by 2.3 PSNR (as detailed in Table 18-(b)). Its reconstruction quality significantly lags behind that of all-one waves, as depicted in the last two columns of Figure 18.

### B.10. Scattering Initial Conditions Spatially

To enhance reconstruction further, we can adjust spatial positions to place the initial conditions, without introducing additional parameters or FLOPs. This concept is straightforward to implement through multi-path FINOLA (refer to Figure 3), where different paths employ scattered initial positions rather than overlapped at

Table 19: **Position of initial conditions.** PSNR values for image reconstruction on the ImageNet-1K validation set is reported. Scattering of initial positions spatially boosts performance.

| Position | #Paths $M$ | #Channels $C$ | PSNR ↑ |
|---|---|---|---|
| Overlapping at Center | 4 | 1024 | 26.1 |
| Scattering Uniformly | 4 | 1024 | **26.7** |
| Overlapping at Center | 8 | 1024 | 27.1 |
| Scattering Uniformly | 8 | 1024 | **28.0** |
| Overlapping at Center | 16 | 1024 | 27.7 |
| Scattering Uniformly | 16 | 1024 | **29.1** |
| Overlapping at Center | 8 | 2048 | 28.0 |
| Scattering Uniformly | 8 | 2048 | **28.9** |
| Overlapping at Center | 16 | 2048 | 29.1 |
| Scattering Uniformly | 16 | 2048 | **30.0** |

the center. Table 19 demonstrates that further improvements in reconstruction is achieved by scattering the initial conditions uniformly compared to placing them at the center, regardless of whether we use 4, 8 or 16 FINOLA paths.

## C. Masked FINOLA for Self-supervised Pre-training

### C.1. Masked FINOLA

FINOLA can be applied to self-supervised learning through a straightforward masked prediction task, which we refer to as *Masked FINOLA* to distinguish it from the vanilla FINOLA. Unlike vanilla FINOLA that support various resolutions of feature map, masked FINOLA performs mask prediction at resolution $\frac{1}{16}$, which is consistent with established baselines like MAE (He et al., 2021), SimMIM (Xie et al., 2022). In this paper, we only use single path for masked FINOLA.

**Simple block masking:** FINOLA is applied through a simple masked prediction design that involves using a single unmasked image block (see Figure 19) to predict the surrounding masked region. Specifically, we crop out the unmasked block and pass it through the encoder, leveraging the power of FINOLA to generate a full-size feature map. Finally, a decoder is applied to recover the pixels in masked region. Unlike vanilla FINOLA, the reconstruction loss is computed only from the masked region. Please note that the unmasked block floats around the image randomly.

Table 20: **Mobile-Former decoder specifications for COCO object detection:** 100 object queries with dimension 256 are used. "down-conv" includes a 3×3 depthwise convolution (stride=2) and a pointwise convolution (256 channels). "up-conv" uses bilinear interpolation, followed by a 3×3 depthwise and a pointwise convolution. "M-F$^+$" replaces the *Mobile* sub-block with a transformer block, while "M-F$^-$" uses the lite bottleneck (Li et al., 2021b) to replace the *Mobile* sub-block.

| Stage | `MF-Dec-522` | | `MF-Dec-211` | |
|---|---|---|---|---|
| query | 100×256 | | 100×256 | |
| $\frac{1}{32}$ | down-conv M-F$^+$ | ×5 | down-conv M-F$^+$ | ×2 |
| $\frac{1}{16}$ | up-conv M-F$^-$ | ×2 | up-conv M-F$^-$ | ×1 |
| $\frac{1}{8}$ | up-conv M-F$^-$ | ×2 | up-conv M-F$^-$ | ×1 |

**Masked FINOLA variants:** Masked FINOLA comprises two variants: the element-wise approach (Masked-FINOLA-E)

Figure 19: **Two Masked FINOLA variants:** element-wise (***left***) and block-wise (***right***) approaches. In the element-wise approach, autoregression is performed similarly to vanilla FINOLA, with the compressed vector ***q*** observing only the unmasked block rather than the entire image. Conversely, the block-wise approach does not compress the unmasked block. Each unmasked position exclusively predicts three masked positions, as indicated by arrows, using Eq. 1. Assignments are grouped together, with shared offsets within each group. The grouping varies depending on the location of the unmasked quadrant, resulting in 1, 2, and 4 groups for corner, edge, and middle locations, respectively. Best viewed in color.

and the block-wise approach (Masked-FINOLA-B), as depicted in Figure 19.

The element-wise variant (Masked-FINOLA-E) operates similarly to vanilla FINOLA, with the compressed vector ***q*** only observing the unmasked block rather than the entire image (see Figure 19-left). To accommodate the longer training required in masked FINOLA (e.g., 1600 epochs), we follow (He et al., 2021) to replace the convolutional decoder with a simple linear layer, transforming a $C$-channel token into a $16{\times}16{\times}3$ image patch.

In contrast, the block-wise variant (Masked-FINOLA-B) preserves the unmasked block in its entirety, without compression. It requires the unmasked block to have a quadrant size. As shown in Figure 19-right, each unmasked position is tasked with predicting three masked positions, denoted by arrows and computed using Eq. 1. These assignments are organized into groups, and within each group, all unmasked positions share common offsets for reaching their assigned masked positions. The configuration of these groups dynamically adapts based on the location of the unmasked quadrant, resulting in 1, 2, or 4 groups for corner, edge, or middle positions, respectively. To promote communication across these groups, transformer blocks are integrated into the decoder.

**Relation to MAE** (He et al., 2021): Masked FINOLA shares a similar architecture with MAE but differs notably in ***masking*** and ***prediction*** strategies. Firstly, masked FINOLA adopts a regular masking design, grouping all unmasked patches into a single block, in contrast to MAE's utilization of random unmasked patches. This design choice suits efficient CNN-based networks. Secondly, masked FINOLA employs a first-order norm+linear autoregression approach for predicting the masked region, whereas MAE utilizes masked tokens within an attention model.

## C.2. Implementation Details

### C.2.1. DECODER ARCHITECTURES

Below, we describe (a) the decoders employed in masked FINOLA, (b) the decoders designed for image classification, and (c) the decoders tailored for object detection.

**Decoders for FINOLA pre-training:** Unlike vanilla FINOLA, which employs stacked upsampling and convolution blocks, the masked FINOLA variants utilize simpler architectures — a linear layer for transforming features into $16{\times}16$ image patches. This choice facilitates longer training. The decoder of Masked-FINOLA-B incorporates transformer blocks (without positional embedding) to enable spatial communication. Masked FINOLA undergoes training for 1600 epochs.

**Decoders for ImageNet classification:** We utilize three decoders to evaluate the pre-trained encoders in FINOLA. These decoders are as follows:

Table 22: **Settings for linear probing and `tran-1` probing on ImageNet-1K:** The encoders are frozen during both tasks.

| Config | Linear probing | `tran-1` probing |
|---|---|---|
| optimizer | SGD | AdamW |
| base learning rate | 0.1 | 0.0005 |
| weight decay | 0 | 0.1 |
| batch size | 4096 | 4096 |
| learning rate schedule | cosine decay | cosine decay |
| warmup epochs | 10 | 10 |
| training epochs | 90 | 200 |
| augmentation | RandomResizeCrop | RandAug (9, 0.5) |
| label smoothing | – | 0.1 |
| dropout | – | 0.1 (MF-W720) 0.2 (MF-W1440/W2880) |
| random erase | – | 0 (MF-W720/W1440) 0.25 (MF-W2880) |

- `lin` decoder: It consists of a single linear layer and is used for linear probing.

- `tran-1` decoder: It incorporates a shallower transformer decoder with a single transformer block followed by a linear classifier and is employed for `tran-1` probing and fine-tuning.

- `tran-4` decoder: This decoder is composed of four transformer blocks followed by a linear classifier and is utilized for fine-tuning alone.

The transformer decoders are designed with different widths (192, 384, 768) to correspond with the three Mobile-Former encoders, which have widths of 720, 1440, and 2880, respectively.

**Decoders for object detection:** The decoders used in the DETR framework with Mobile-Former (Chen et al., 2022) are described in Table 20. Both decoders consist of 100 object queries with a dimension of 256. While they share a similar structure across three scales, they differ in terms of their depths. Since the backbone network ends at a resolution of $\frac{1}{16}$, the decoder incorporates a downsampling step to further reduce the resolution to $\frac{1}{32}$. This enables the decoder to efficiently process the features for object detection.

### C.2.2. TRAINING SETUP

In this section, we provide detailed training setups for different tasks, including:

- Masked FINOLA pre-training on ImageNet-1K.

- Linear probing on ImageNet-1K.

- `tran-1` probing on ImageNet-1K.

- Fine-tuning on ImageNet-1K.

- COCO object detection.

**Masked FINOLA pre-training:** Similar to the vanilla FINOLA, masked FINOLA also follows the training setup described in Table 21, but with a larger batch size due to the simpler decoder architecture that requires less memory consumption.

Table 21: **Pre-training setting for masked FINOLA.**

| Config | Masked FINOLA |
|---|---|
| optimizer | AdamW |
| base learning rate | 1.5e-4 |
| weight decay | 0.1 |
| batch size | 1024 |
| learning rate schedule | cosine decay |
| warmup epochs | 10 |
| training epochs | 1600 |
| image size | $256^2$ |
| augmentation | RandomResizeCrop |

**Linear probing:** In our linear probing, we follow the approach described in (He et al., 2021) by incorporating an additional BatchNorm layer without affine transformation (affine=False). Detailed settings can be found in Table 22.

Table 23: **Setting for end-to-end fine-tuning on ImageNet-1K.**

| Config | Value |
|---|---|
| optimizer | AdamW |
| base learning rate | 0.0005 |
| weight decay | 0.05 |
| layer-wise lr decay | 0.90 (MF-W720/W1440) 0.85 (MF-W2880) |
| batch size | 512 |
| learning rate schedule | cosine decay |
| warmup epochs | 5 |
| training epochs | 200 (MF-W720) 150 (MF-W1440) 100 (MF-W2880) |
| augmentation | RandAug (9, 0.5) |
| label smoothing | 0.1 |
| mixup | 0 (MF-W720) 0.2 (MF-W1440) 0.8 (MF-W2880) |
| cutmix | 0 (MF-W720) 0.25 (MF-W1440) 1.0 (MF-W2880) |
| dropout | 0.2 |
| random erase | 0.25 |

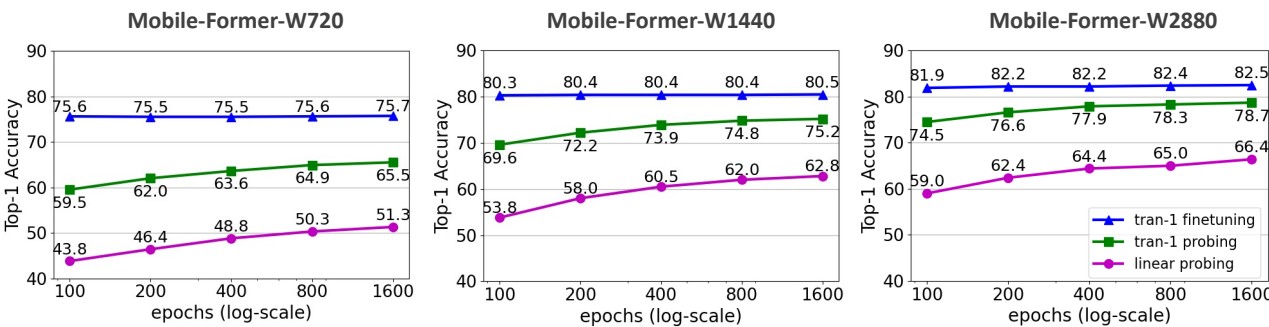

Figure 20: **Training schedules of Masked-FINOLA-B.** Longer training schedule provides consistent improvement for linear and `tran-1` probing over different models, while fine-tuning performance is not sensitive to training schedule. Best viewed in color.

**`tran-1` probing:** The settings for `tran-1` decoder probing are presented in Table 22. It is important to note that the default decoder widths are 192, 384, and 768 for MF-W720, MF-W1440, and MF-W2880, respectively.

**End-to-end fine-tuning on ImageNet-1K:** The settings for the end-to-end fine-tuning of both the encoder and `tran-1` decoder are presented in Table 23. The decoder weights are initialized from the `tran-1` probing stage.

**Decoder probing on COCO object detection:** In this configuration, the backbone pre-trained on ImageNet-1K is frozen, and only the decoders are trained for 500 epochs on 8 GPUs with 2 images per GPU. We employ AdamW optimizer with an initial learning rate of 1e-4. The learning rate is decreased by a factor of 10 after 400 epochs. The weight decay is 1e-4, and the dropout rate is 0.1.

**Fine-tuning on COCO object detection:** In this setting, both the encoder and decoder are fine-tuned. The fine-tuning process consists of an additional 200 epochs following the decoder probing stage. The initial learning rate for both the encoder and decoder is set to 1e-5, which decreases to 1e-6 after 150 epochs.

### C.3. Ablation Studies

**Ablation on training schedule:** The impact of training schedule length on three Mobile-Former encoders is depicted in Figure 20. Notably, the accuracies of both linear and `tran-1` probings demonstrate a consistent improvement as the training duration increases. Interestingly, even with a pre-training of just 100 epochs, fine-tuning with `tran-1` achieves commendable performance. This finding diverges from the observations in MAE (He et al., 2021), where longer training is

Table 24: **Ablation on the number of transformer blocks in the decoder:** Evaluation is conducted on ImageNet using Mobile-Former-W2880 as the encoder. Each transformer block consists of 512 channels. Each model is pre-trained for 800 epochs. Increasing the decoder depth exhibits consistent improvement for linear and `tran-1` probing, while fine-tuning performance shows limited sensitivity to decoder depth.

| #Blocks | lin | tran-1 | tran-1-ft |
|---------|------|--------|-----------|
| 1 | 61.1 | 74.4 | 82.2 |
| 2 | 62.6 | 76.5 | 82.3 |
| 3 | 63.5 | 77.3 | 82.2 |
| 4 | 63.8 | 78.0 | 82.3 |
| 5 | 64.0 | 78.1 | 82.3 |
| 6 | 65.0 | 78.3 | 82.4 |

Table 25: **Comparison with masked encoding methods on ImageNet-1K using linear probing**. The baseline methods include iGPT (Chen et al., 2020a), BEiT (Bao et al., 2021), SimMIM (Xie et al., 2022), MAE (He et al., 2021) and MAE-Lite (Wang et al., 2022). Three Mobile-Former backbones of varying widths are used. FINOLA pre-training demonstrates the ability to learn effective representations for small models. [†] denotes our implementation.

| Method | Model | Params | Top-1 |
|--------|-------|--------|-------|
| iGPT | iGPT-L | 1362M | 69.0 |
| BEiT | ViT-B | 86M | 56.7 |
| SimMIM | ViT-B | 86M | 56.7 |
| MAE | ViT-B | 86M | 68.0 |
| MAE[†] | ViT-S | 22M | 49.2 |
| MAE-Lite | ViT-Tiny | 6M | 23.3 |
| **FINOLA** | MF-W720 | 6M | 51.3 |
| **FINOLA** | MF-W1440 | 14M | 62.8 |
| **FINOLA** | MF-W2880 | 28M | 66.4 |

essential for fine-tuning improvements.

**Ablation on the number of transformer blocks in the decoder:** We investigate the impact of the number of transformer blocks in the decoder on FINOLA pre-training using the Mobile-Former-W2880 as encoder. Each transformer block in the decoder consists of 512 channels, but does *not* use positional embedding. The results, shown in Table 24, demonstrate that adding more transformer blocks leads to consistent improvements in both linear and `tran-1` probing tasks. However, we observe that the performance of fine-tuning is less sensitive to changes in the decoder depth.

### C.4. Comparable Performance with Established Baselines on Linear Probing

As shown in Table 25, FINOLA achieves comparable performance with well known baselines on linear probing while requiring lower FLOPs. The comparison is conducted in end-to-end manner (combining encoder and pre-training method). For example, we compare FINOLA+MobileFormer with MAE+ViT in the context of ImageNet classification.

### C.5. Robust Task Agnostic Encoders

**FINOLA provides a robust task-agnostic encoders:** Pre-training with FINOLA followed by fine-tuning on ImageNet-1K (IN-1K) consistently outperforms IN-1K supervised pre-training in both ImageNet classification and COCO object detection (see Figure 6). The gains in object detection are substantial, ranging from 5 to 6.4 AP. Remarkably, even without IN-1K fine-tuning, FINOLA pre-training alone outperforms the supervised counterpart in object detection by a clear margin (3 to 4.5 AP). This highlights FINOLA's ability to encode spatial structures.

**Comparisons with MoCo-v2**: As shown in Table 26, FINOLA demonstrates comparable performance to MoCo-V2 in linear probing, while surpassing MoCo-V2 in `tran-1` probing that uses a single transformer block as a decoder for classification, IN-1K fine-tuning, object detection and segmentation. The backbone is frozen for both COCO object detection and segmentation. FINOLA's superior performance suggests it learns more effective intermediate features, contributing to more representative decoder features. Furthermore, the improved performance in object detection emphasizes FINOLA's ability to encode spatial structures effectively.

These experiments demonstrate that the proposed masked FINOLA is able to learn task-agnostic representation by using a simple masking design. This supports that the underling PDEs capture the intrinsic spatial structures present in images.

**Comparison with the IN-1K supervised pre-training on transferring to COCO object detection:** Table 27 presents the results of COCO object detection using frozen backbones. The evaluation utilizes three Mobile-Former encoders with different widths and two Mobile-Former decoders with different depths. Notably, FINOLA pre-training followed

Table 26: **Comparisons with MoCo-v2 (Chen et al., 2020d)** on ImageNet classification, COCO object detection and instance segmentation. Three Mobile-Former backbones with different widths are used. In `tran-1`, the encoder is frozen while a transformer block is trained as a decoder using class labels. In `tran-1-ft`, encoders are fine-tuned. Encoders are frozen in both COCO object detection and instance segmentation. DETR framework is used for object detection, while Mask-RCNN ($1\times$) is used for segmentation. FINOLA outperforms MoCo-V2 in most evaluations, except on par in linear probing.

| Pre-training | Encoder | IN-1K Top-1 | | | COCO Det (Box-AP) | | COCO Seg (Mask-AP) | |
| --- | --- | --- | --- | --- | --- | --- | --- | --- |
| | | lin | tran-1 | tran-1-ft | w/o IN-ft | with IN-ft | w/o IN-ft | with IN-ft |
| MoCo-V2 | MF-W720 | **51.6** | 52.9 | 74.3 | 31.8 | 39.9 | 23.2 | 25.3 |
| **FINOLA** | | 51.3 | **65.5** | **75.6** | **40.0** | **41.6** | **26.3** | **28.4** |
| MoCo-V2 | MF-W1440 | 60.4 | 58.5 | 79.2 | 30.3 | 39.0 | 25.6 | 25.7 |
| **FINOLA** | | **62.8** | **75.2** | **80.5** | **42.6** | **44.0** | **30.6** | **32.7** |
| MoCo-V2 | MF-W2880 | **66.5** | 63.8 | 80.0 | 25.5 | 31.7 | 27.8 | 25.2 |
| **FINOLA** | | 66.4 | **78.7** | **82.5** | **43.3** | **45.5** | **33.3** | **35.1** |

Table 27: **COCO object detection results** on the `val2017` dataset using a ***frozen*** backbone pre-trained on ImageNet-1K. Evaluation is conducted over three backbones and two heads that use Mobile-Former (Chen et al., 2022) end-to-end in DETR (Carion et al., 2020) framework. Our FINOLA consistently outperform the supervised counterpart. Notably, fine-tuning on ImageNet-1K (denoted as "IN-ft") yields further improvements. The initial "MF" (e.g., `MF-Dec-522`) denotes Mobile-Former. The madds metric is based on an image size of $800\times1333$.

| | Head | | | Backbone | | | | | | | | | | |
| --- | --- | --- | --- | --- | --- | --- | --- | --- | --- | --- | --- | --- | --- | --- |
| model | madds (G) | param (M) | | model | madds (G) | param (M) | pre-train | IN-ft | AP | AP$_{50}$ | AP$_{75}$ | AP$_S$ | AP$_M$ | AP$_L$ |
| MF Dec 522 | 34.6 | 19.4 | | MF W2880 | 77.5 | 25.0 | supervised | – | 40.5 | 58.5 | 43.3 | 21.1 | 43.4 | 56.8 |
| | | | | | | | **FINOLA** | ✗ | 43.3 $_{(+2.8)}$ | 61.5 | 46.8 | 23.7 | 46.9 | 60.1 |
| | | | | | | | **FINOLA** | ✓ | **45.5** $_{(+5.0)}$ | **63.8** | **49.5** | **25.1** | **49.1** | **63.5** |
| | 32.3 | 18.6 | | MF W1440 | 20.4 | 11.7 | supervised | – | 38.3 | 56.0 | 40.8 | 19.0 | 40.9 | 54.3 |
| | | | | | | | **FINOLA** | ✗ | 42.6 $_{(+4.3)}$ | 60.3 | 46.1 | 22.6 | 46.2 | 60.0 |
| | | | | | | | **FINOLA** | ✓ | **44.0** $_{(+5.7)}$ | **62.3** | **47.3** | **23.8** | **47.6** | **61.0** |
| | 31.1 | 18.2 | | MF W720 | 5.6 | 4.9 | supervised | – | 35.2 | 52.1 | 37.6 | 16.9 | 37.2 | 51.7 |
| | | | | | | | **FINOLA** | ✗ | 40.0 $_{(+4.8)}$ | 57.9 | 42.9 | 20.6 | 43.3 | 56.8 |
| | | | | | | | **FINOLA** | ✓ | **41.6** $_{(+6.4)}$ | **59.4** | **45.0** | **21.2** | **45.0** | **58.9** |
| MF Dec 211 | 15.7 | 9.2 | | MF W2880 | 77.5 | 25.0 | supervised | – | 34.1 | 51.3 | 36.1 | 15.5 | 36.8 | 50.0 |
| | | | | | | | **FINOLA** | ✗ | 36.7 $_{(+2.6)}$ | 53.7 | 39.3 | 18.2 | 39.7 | 52.2 |
| | | | | | | | **FINOLA** | ✓ | **41.0** $_{(+6.9)}$ | **59.2** | **44.4** | **20.9** | **44.6** | **58.3** |
| | 13.4 | 8.4 | | MF W1440 | 20.4 | 11.7 | supervised | – | 31.2 | 47.8 | 32.8 | 13.7 | 32.9 | 46.9 |
| | | | | | | | **FINOLA** | ✗ | 36.0 $_{(+4.8)}$ | 52.7 | 38.7 | 16.6 | 39.1 | 52.5 |
| | | | | | | | **FINOLA** | ✓ | **39.2** $_{(+8.0)}$ | **56.9** | **42.0** | **19.7** | **42.8** | **56.2** |
| | 12.2 | 8.0 | | MF W720 | 5.6 | 4.9 | supervised | – | 27.8 | 43.4 | 28.9 | 11.3 | 29.1 | 41.6 |
| | | | | | | | **FINOLA** | ✗ | 33.0 $_{(+5.2)}$ | 49.3 | 35.0 | 15.3 | 35.1 | 48.9 |
| | | | | | | | **FINOLA** | ✓ | **35.8** $_{(+8.0)}$ | **52.6** | **38.3** | **16.4** | **38.3** | **52.0** |

by ImageNet-1K (IN-1K) fine-tuning consistently outperforms the IN-1K supervised pre-training across all evaluations, demonstrating the effectiveness of task-agnostic encoders. Impressively, even FINOLA pre-training alone, without IN-1K fine-tuning, surpasses the supervised counterpart on object detection by a significant margin of 2.6–5.2 AP. This showcases FINOLA's ability to encode spatial structures.

### C.6. Fine-tuning on COCO

Furthermore, fine-tuning the backbone on COCO further enhances detection performance. Table 28 provides a comprehensive comparison of fine-tuning results using the Mobile-Former (Chen et al., 2022) in the DETR (Carion et al., 2020) framework. Unlike the frozen backbone configuration, where FINOLA outperforms supervised pre-training significantly (as shown in

Table 28: **COCO object detection results** on the `val2017` dataset after ***fine-tuning*** both the backbone and head on COCO. Evaluation is performed on three different backbones and two heads, utilizing the Mobile-Former (Chen et al., 2022) end-to-end in the DETR (Carion et al., 2020) framework. Our approach, which involves FINOLA pre-training followed by ImageNet-1K fine-tuning, surpasses the performance of the supervised baselines. The initial "MF" (e.g., `MF-Dec-522`) denotes Mobile-Former, while "IN-ft" indicates fine-tuning on ImageNet-1K. The reported madds values are based on the image size of $800 \times 1333$.

| model | Head madds (G) | param (M) | model | Backbone madds (G) | param (M) | pre-train | IN-ft | AP | AP$_{50}$ | AP$_{75}$ | AP$_S$ | AP$_M$ | AP$_L$ |
|---|---|---|---|---|---|---|---|---|---|---|---|---|---|
| MF Dec 522 | 34.6 | 19.4 | MF W2880 | 77.5 | 25.0 | supervised | – | 48.1 | 66.6 | 52.5 | 29.7 | 51.8 | 64.0 |
| | | | | | | FINOLA | ✗ | 48.0$_{(-0.1)}$ | 66.2 | 52.3 | 28.2 | 51.4 | 64.1 |
| | | | | | | FINOLA | ✓ | **49.0** $_{(+0.9)}$ | **67.7** | **53.4** | **30.1** | **52.9** | **65.5** |
| | 32.3 | 18.6 | MF W1440 | 20.4 | 11.7 | supervised | – | 46.2 | 64.4 | 50.1 | 27.1 | 49.8 | 62.4 |
| | | | | | | FINOLA | ✗ | 46.8$_{(+0.6)}$ | 64.9 | 51.0 | 26.6 | 50.6 | 63.4 |
| | | | | | | FINOLA | ✓ | **47.3**$_{(+1.1)}$ | **65.6** | **51.4** | **27.3** | **50.7** | **63.9** |
| | 31.1 | 18.2 | MF W720 | 5.6 | 4.9 | supervised | – | 42.5 | 60.4 | 46.0 | 23.9 | 46.0 | 58.5 |
| | | | | | | FINOLA | ✗ | 43.3$_{(+0.8)}$ | 61.0 | 47.0 | 23.1 | 46.6 | 61.0 |
| | | | | | | FINOLA | ✓ | **44.4**$_{(+1.9)}$ | **62.1** | **48.1** | **24.3** | **47.8** | **61.5** |
| MF Dec 211 | 15.7 | 9.2 | MF W2880 | 77.5 | 25.0 | supervised | – | 44.0 | 62.8 | 47.7 | 25.8 | 47.3 | 60.7 |
| | | | | | | FINOLA | ✗ | 44.4$_{(+0.4)}$ | 62.5 | 48.2 | 24.7 | 47.6 | 60.7 |
| | | | | | | FINOLA | ✓ | **46.0**$_{(+2.0)}$ | **64.8** | **49.9** | **26.2** | **50.0** | **62.7** |
| | 13.4 | 8.4 | MF W1440 | 20.4 | 11.7 | supervised | – | 42.5 | 60.6 | 46.0 | 23.6 | 45.9 | 57.9 |
| | | | | | | FINOLA | ✗ | 42.4$_{(-0.1)}$ | 60.2 | 45.9 | 21.9 | 45.7 | 60.0 |
| | | | | | | FINOLA | ✓ | **43.8**$_{(+1.3)}$ | **61.8** | **47.5** | **23.9** | **47.1** | **60.8** |
| | 12.2 | 8.0 | MF W720 | 5.6 | 4.9 | supervised | – | 37.6 | 55.1 | 40.4 | 18.9 | 40.6 | 53.8 |
| | | | | | | FINOLA | ✗ | 37.2$_{(-0.4)}$ | 54.3 | 39.7 | 18.7 | 39.8 | 53.4 |
| | | | | | | FINOLA | ✓ | **39.3**$_{(+1.7)}$ | **56.7** | **42.4** | **19.4** | **42.1** | **56.5** |

Table 29: **Comparison with DETR-based models on COCO detection.** All baselines are fine-tuned on COCO. FINOLA-DETR utilizes Mobile-Former (MF-W2880) as the backbone, which has similar FLOPs and model size to the ResNet-50 used in other methods. MAdds are calculated based on an image size of $800 \times 1333$.

| Model | Query | AP | AP$_{50}$ | AP$_{75}$ | AP$_S$ | AP$_M$ | AP$_L$ | MAdds (G) | Param (M) |
|---|---|---|---|---|---|---|---|---|---|
| DETR-DC5(Carion et al., 2020) | **100** | 43.3 | 63.1 | 45.9 | 22.5 | 47.3 | 61.1 | 187 | 41 |
| Deform-DETR(Zhu et al., 2020) | 300 | 46.2 | 65.2 | 50.0 | 28.8 | 49.2 | 61.7 | 173 | 40 |
| DAB-DETR(Liu et al., 2022) | 900 | 46.9 | 66.0 | 50.8 | 30.1 | 50.4 | 62.5 | 195 | 48 |
| DN-DETR(Li et al., 2022a) | 900 | 48.6 | 67.4 | 52.7 | 31.0 | 52.0 | 63.7 | 195 | 48 |
| DINO(Zhang et al., 2022) | 900 | **50.9** | **69.0** | **55.3** | **34.6** | **54.1** | 64.6 | 279 | 47 |
| **FINOLA-DETR (frozen)** | **100** | 45.5 | 63.8 | 49.5 | 25.1 | 49.1 | 63.5 | **112** | **44** |
| **FINOLA-DETR (fine-tune)** | | 49.0 | 67.7 | 53.4 | 30.1 | 52.9 | **65.5** | | |

Table 27), they achieve similar performance in COCO fine-tuning. This is because the advantage of FINOLA pre-training on spatial representation diminishes when object labels in COCO provide strong guidance. However, FINOLA maintains its leading position by leveraging fine-tuning on IN-1K to improve semantic representation and transfer it to object detection. Compared to the supervised baseline, FINOLA pre-training followed by IN-1K fine-tuning achieves a gain of 0.9–2.0 AP for all three encoders and two decoders.

Table 29 compares FINOLA-DETR (in which the backbone is fine-tuned in the DETR framework) with existed DETR baselines. FINOLA-DETR achieves an AP of 49.0, outperforming most DETR-based detectors except DINO (Zhang et al., 2022). Remarkably, our method achieves these results while using significantly fewer FLOPs (112G vs. 279G) and object queries (100 vs. 900). When compared to DETR-DC5 with a fine-tuned backbone, FINOLA-DETR with a *frozen* backbone achieves a 2.2 AP improvement while reducing MAdds by 40%.

Table 30: **Comparing FINOLA and Masked FINOLA** on ImageNet-1K. Masked FINOLA variants trade restoration accuracy for enhanced semantic representation. The block-wise masked FINOLA outperforms the element-wise variant in linear probing (`lin`), probing with a single transformer block (`tran-1`), and fine-tuning (`tran-1-ft`).

| Model | Compress | Autoregression | Decoder | Recon-PSNR | lin | tran-1 | tran-1-ft |
|---|---|---|---|---|---|---|---|
| FINOLA | ✓ | element | up+conv | **25.8** | 17.9 | 46.8 | 81.9 |
| Masked FINOLA-E | ✓ | element | linear | 16.7 | 54.1 | 67.8 | 82.2 |
| Masked FINOLA-B | ✗ | block | trans+linear | 17.3 | **66.4** | **78.7** | **82.5** |

Table 31: **Comparison between FINOLA and Masked FINOLA** on ImageNet (Deng et al., 2009) classification: Compared to masked FI-NOLA variants, FINOLA performs poorly on both linear probing (`lin`) and probing with a single transformer block (`tran-1`) with clear margins. Even we search over the dimension of latent space from 64 to 3072, the gap is still large, i.e. more than 20%. Block-wise masked FINOLA (Masked-FINOLA-B) outperforms the element-wise variant (Masked-FINOLA-E), achieving higher accuracy. Please note that the encoders are frozen when performing linear and `tran-1` probing.

| Pre-training | Dim of $q$ | lin | tran-1 |
|---|---|---|---|
| FINOLA | 64 | 10.2 | 20.2 |
| | 128 | 11.5 | 24.0 |
| | 256 | 15.0 | 29.0 |
| | 512 | 20.1 | 34.1 |
| | 1024 | 23.0 | 39.6 |
| | 2048 | 23.2 | 41.1 |
| | 3072 | 17.9 | 46.8 |
| Masked FINOLA-E | 512 | 54.1 | 67.8 |
| Masked FINOLA-B | —— | 66.4 | 78.7 |

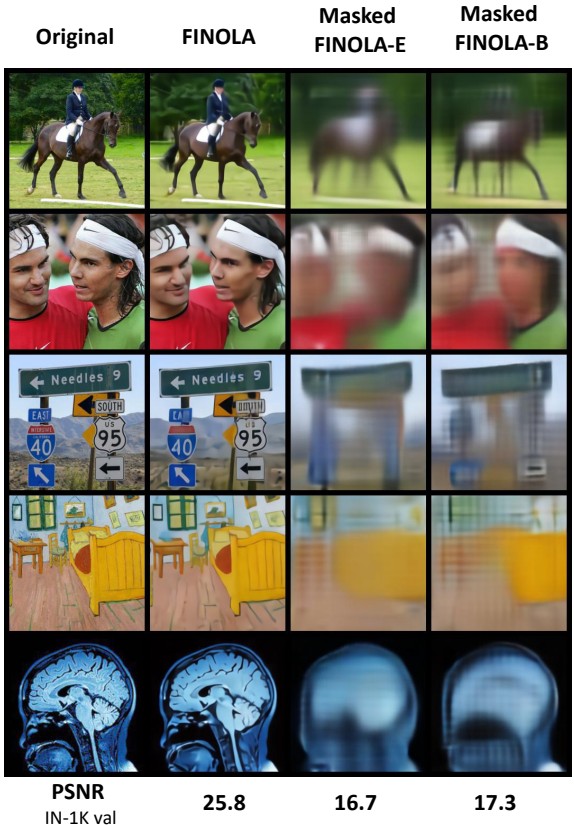

Figure 21: **FINOLA vs. masked FINOLA on image reconstruction:** In this comparison, the encoders of the two masked FINOLA variants are frozen, and their attentional pooling and FINOLA components are fine-tuned. To ensure a fair comparison, we replace the decoders in the masked FINOLA variants with the same architecture as FINOLA, trained from scratch. When compared to vanilla FINOLA, the masked variants preserve color and shape information but exhibit a loss of texture details.

These results showcase the efficacy of FINOLA in capturing rich image representations even with more compact models, offering a promising approach for efficient self-supervised learning.

# D. Comparison between FINOLA and Masked FINOLA

## D.1. Detailed Experimental Results

Table 30 presents a comparison between vanilla FINOLA and two masked FINOLA variants, assessing both their architectural distinctions and performance in image reconstruction and classification tasks. The introduction of masking, a characteristic

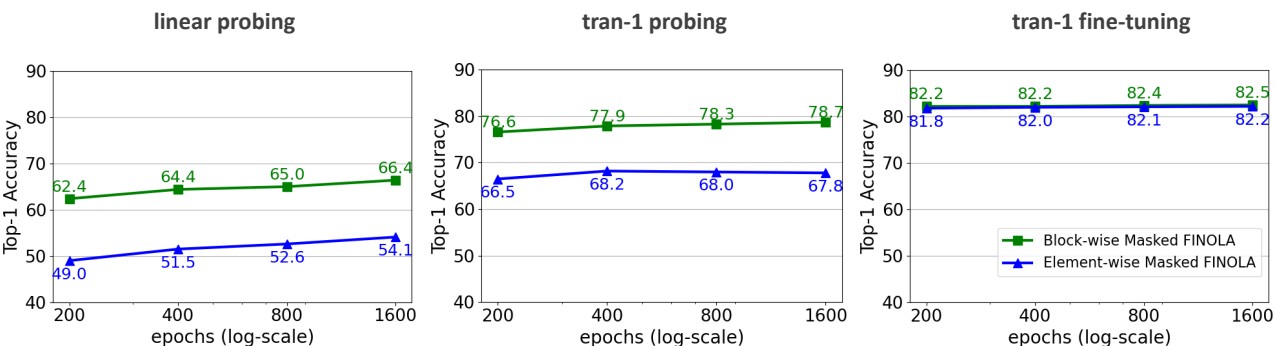

Figure 22: **Comparison of element-wise and block-wise Masked FINOLA**. The evaluation includes linear probing, `tran-1` probing, and `tran-1` fine-tuning. Block-wise masked FINOLA consistently outperforms the element-wise counterpart across all evaluations. Notably, the performance gap in fine-tuning is smaller compared to linear and `tran-1` probing. Best viewed in color.

of masked FINOLA, entails a trade-off between restoration accuracy and enhanced semantic representation.

**Comparison of FINOLA and Masked FINOLA on ImageNet classification:** Table 31 presents the results of linear and `tran-1` probing applied to the vanilla FINOLA across various dimensions of the latent space. Notably, even the highest accuracy achieved by the vanilla FINOLA falls significantly behind both masked FINOLA variants (element-wise or block-wise). This stark difference highlights the remarkable power of masked prediction in learning semantic representations.

**Comparison of FINOLA and Masked FINOLA on image reconstruction:** Figure 21 presents a comparison of re-constructed samples obtained using FINOLA and masked FINOLA. In the case of the two masked FINOLA variants (element-wise and block-wise), the encoders are frozen, and only their attentional pooling and FINOLA components are fine-tuned. To ensure a fair comparison, we utilize the same architecture for the decoders in the masked FINOLA variants as in FINOLA, training them from scratch. The corresponding peak signal-to-noise ratio (PSNR) values on the ImageNet validation set are provided at the bottom. While the masked variants preserve color and shape information, they exhibit a loss of texture details compared to the vanilla FINOLA. Notably, as demonstrated in the main paper, the masked FINOLA variants demonstrate stronger semantic representation. This comparison highlights that FINOLA and masked FINOLA adhere to the same mathematical principles (involving partial differential equations) but strike different balances between semantic representation and preserving fine details.

**Comparison between two Masked FINOLA variants:** Figure 22 showcases the results of linear probing, `tran-1` probing, and fine-tuning for two masked FINOLA variants trained with different schedules. The block-wise masked FINOLA consistently outperforms its element-wise counterpart across all evaluations. These findings demonstrate the effectiveness of directly applying FINOLA on the unmasked features to predict the masked region, as opposed to performing compression before applying FINOLA.

### D.2. Geometric Insight

Geometrically, Figure 23 illustrates masked FINOLA introduces a substantial increase in Gaussian curvature on critical feature surfaces, suggesting enhanced curvature in the latent space for capturing semantics.

### D.3. Calculation of Gaussian Curvature

To compute the Gaussian curvature, we consider the feature map per channel as a set of $W \times H$ surfaces $z_k(x, y)$ in 3D space, where $x$, $y$, and $z_k$ denote the coordinates. At each position $(x, y)$, the Gaussian curvature for the $k^{th}$ channel can be determined using the following equation:

$$\kappa_k(x, y) = \frac{\frac{\partial^2 z_k}{\partial x^2} \frac{\partial^2 z_k}{\partial y^2} - \left(\frac{\partial^2 z_k}{\partial x \partial y}\right)^2}{\left(1 + \left(\frac{\partial z_k}{\partial x}\right)^2 + \left(\frac{\partial z_k}{\partial y}\right)^2\right)^2}. \tag{10}$$

Figure 23: **FINOLA vs. Masked FINOLA on Gaussian curvature of critical features**. Masked FINOLA demonstrates significantly larger curvature on critical features than vanilla FINOLA, highlighting the effectiveness of masked prediction in curving the latent space to capture semantics. Best viewed in color.

Gaussian curvature is computed for all channels at each grid element. Subsequently, channels within each image are sorted based on the root mean square of the peak positive curvature ($\kappa_+$) and the peak negative curvature ($\kappa_-$) over the surface.

## E. Mathematical Derivation

### E.1. Normalization after Diagonalization

Below, we provide the derivation of $\hat{\psi}_i$ in Eq. 7.

$$\hat{\psi}_i = \frac{(C\boldsymbol{I} - \boldsymbol{J})\boldsymbol{V}\psi_i}{\sqrt{\psi_i^T \boldsymbol{V}^T (C\boldsymbol{I} - \boldsymbol{J})\boldsymbol{V}\psi_i}}. \tag{11}$$

A FINOLA path is described as $\Delta_x \phi_i = \boldsymbol{A}\hat{\phi}_i$ (Eq. 3 in the paper), where $\hat{\phi}_i$ is a normalized $\phi_i$, i.e. $\phi_i = \frac{\phi_i - \mu}{\sigma}$. After diagonalization of $\boldsymbol{A}\boldsymbol{B}^{-1} = \boldsymbol{V}\boldsymbol{\Lambda}\boldsymbol{V}^{-1}$, the FINOLA vectors $\phi_i$ are projected into $\psi_i = \boldsymbol{V}^{-1}\phi_i$, where each $\psi_i$ satisfies a one-way wave equation.

We attempt to rewrite $\psi_i$ in the FINOLA format as $\Delta_x \psi_i = \boldsymbol{H}_A \hat{\psi}_i$ (similar to $\phi_i$ before projection $\Delta_x \phi_i = \boldsymbol{A}\hat{\phi}_i$). $\hat{\psi}_i$ is not a simple normalization. The derivation of $\boldsymbol{H}_A$ and $\hat{\psi}_i$ is shown below step by step:

$$
\begin{aligned}
\Delta_x \psi_i &= \boldsymbol{V}^{-1}\Delta_x \phi_i && (\psi_i = \boldsymbol{V}^{-1}\phi_i) \\
&= \boldsymbol{V}^{-1}\boldsymbol{A}\hat{\phi}_i = \boldsymbol{V}^{-1}\boldsymbol{A}\frac{\phi_i - \mu}{\sigma} && (\Delta_x \phi_i = \boldsymbol{A}\hat{\phi}_i) \\
&= \boldsymbol{V}^{-1}\boldsymbol{A}\frac{\phi_i - \frac{1}{C}\boldsymbol{J}\phi_i}{\sqrt{\frac{1}{C}\phi_i^T \phi_i - \frac{1}{C^2}\phi_i^T \boldsymbol{J}\phi_i}} && (\mu, \sigma \text{ in matrix format}, \boldsymbol{J} \text{ is all one matrix}) \\
&= \boldsymbol{V}^{-1}\boldsymbol{A}\frac{(C\boldsymbol{I} - \boldsymbol{J})\phi_i}{\sqrt{\phi_i^T (C\boldsymbol{I} - \boldsymbol{J})\phi_i}} \\
&= \boldsymbol{V}^{-1}\boldsymbol{A}\frac{(C\boldsymbol{I} - \boldsymbol{J})\boldsymbol{V}\psi_i}{\sqrt{\psi_i^T \boldsymbol{V}^T (C\boldsymbol{I} - \boldsymbol{J})\boldsymbol{V}\psi_i}} && (\phi_i = \boldsymbol{V}\psi_i)
\end{aligned} \tag{12}
$$

Thus, we have:

$$\boldsymbol{H}_A = \boldsymbol{V}^{-1}\boldsymbol{A}, \qquad \hat{\psi}_i = \frac{(C\boldsymbol{I} - \boldsymbol{J})\boldsymbol{V}\psi_i}{\sqrt{\psi_i^T \boldsymbol{V}^T (C\boldsymbol{I} - \boldsymbol{J})\boldsymbol{V}\psi_i}}. \tag{13}$$

