# OpenReview forum: "Exploring Invariance in Images through One-way Wave Equations"
_ICML.cc/2025/Conference — ICML 2025 poster_

### Official Review · Reviewer_D8KQ · 2025-03-13

**Overall Recommendation:** 3

**Summary:**

This paper draws connections between recurring regression using first-order norm + linear autoregression and the discretized one-way wave equation. Through this framework the paper proposes a model to embed images and provides empirical evidence supporting the model performance for image reconstruction. Moreover, the paper provides supporting evidence that this model is more memory efficient than existing methods and provides improved speed under parallelization.

## update after rebuttal

In my opinion, the rebuttal has done a great job in the following aspects:
- The acknowledgement of overstatements and willingness to revise these statements
- Providing additional analysis on the diagonalizability and invertibility of the matrices key to their approach.

Given the good PSNR and poor FID scores along with a lack of theoretical connections, it remains unclear if traveling waves reveal a promising new avenue for autoregression on images. However, the paper makes a good case of empirical evidence supporting this approach.

**Claims And Evidence:**

The central claim of the paper is the following: *Images share a set of one-way wave equations in the latent feature space*. This is evidenced through empirical comparison of on peak signal-to-noise ratio on the ImageNet and Kodak datasets. However, **this strong statement lacks strong theoretical support.** Moreover, simple image regression models achieve high PSNR by blurring the image which appears to be the evidence illustrated in Figure 2 as well as the reconstructions in the appendix. Without additional standard metrics (FID/IS/SSIM) for comparison, these empirical results are also insufficient to support the papers central claim.

In addition, the paper makes claims that this model is computationally more efficient in both speed and memory than existing methods. This is evidenced by comparing the dimension of the latent state, number of latent parameters and the bits per pixel across several datasets. This paper makes a claim that the model is capable of providing computational speed ups using parallelization. This is evidenced by inference time comparisons on a single image.

**Essential References Not Discussed:**

The paper thoroughly explores the use and impact of autoregressive models, but under explores of the importance or impact of traveling waves in neural networks and image processing. See above for some example references.

**Experimental Designs Or Analyses:**

I did not verify the experimental analysis as I did not find any supplementary material provided.

**Methods And Evaluation Criteria:**

The proposed evaluation criteria and methods make sense and are necessary for evaluating the current model. However, they are insufficient in supporting the papers claim.

**Other Comments Or Suggestions:**

Some typos like line one of the abstract: *we empirically reveals an invariance over images*.

**Other Strengths And Weaknesses:**

**Strengths:** To the best of my knowledge the derivation of the one-way wave equation from FINOLA is a novel derivation. In addition there are a substantial number of tasks and results presented.

**Weaknesses:** Strong central claim without appropriate theoretical support. Insufficient metrics to adequately determine the full performance of the architecture.

**Questions For Authors:**

Can you provide more detailed theoretical insights or formal proofs that justify the assumption that images naturally conform to a one-way wave equation in the latent feature space?

Beyond PSNR, have you evaluated the model using perceptually motivated metrics (e.g., SSIM, FID, or IS)?

Is it possible to report the wave speed itself? Have you considered a framework where the wave speed varies spatially or by image rather than being invariant?

Given that the derivation relies on assumptions such as the invertibility and diagonalizability of weight matrices, how sensitive is the model’s performance to these assumptions? Specifically, have you examined the condition numbers of matrices B or Q in practice, and what impact do they have on the stability and performance of the model?

**Relation To Broader Scientific Literature:**

This work demonstrating a network which admits a latent traveling wave solution mirrors some related works which demonstrate the capacity to generate traveling wave solutions in the latent state of neural networks [1]. This paper offers some novelty by using the FINOLA method and discretizing solely about the spatial differences. However, the central claim that all images should admit a common spatial wave-equation with spatially invariant weights appears at odds with recent techniques which perform image segmentation with traveling waves and demonstrate that unique spatiotemporal patterns emerge for each object in an input image [2].

[1] Keller, T. Anderson, et al. _TRAVELING WAVES ENCODE THE RECENT PAST AND ENHANCE SEQUENCE LEARNING_. 2024.

[2] Liboni, Luisa H. B., et al. _Image Segmentation with Traveling Waves in an Exactly Solvable Recurrent Neural Network_. arXiv:2311.16943, arXiv, 28 Nov. 2023. _arXiv.org_, [https://doi.org/10.48550/arXiv.2311.16943](https://doi.org/10.48550/arXiv.2311.16943).

**Theoretical Claims:**

I have checked the derivation of the one-way wave equation from the FINOLA architecture. This derivation relies on several key assumptions such as the invertibility and diagonalizability of general weight matrices. As a note, these derivations lack some small but important details such as the commutativity of $V^{-1}$ with $\nabla_x,\nabla_y$,. The larger issue is that this derivation demonstrates a property of the network, that it learns a fixed wave speed, rather than a property of images. This would be significantly improved if the claim were relaxed or proper theoretical propositions were provided about the dynamics of the underlying latent state.

---

> ### Author Rebuttal · Authors · 2025-04-01
>
> We are grateful for the reviewer's valuable feedback.
>
> $\color{blue}{[Question-1]:}$
> **Can you provide more detailed theoretical insights or formal proofs that justify the assumption that images naturally conform to a one-way wave equation in the latent feature space?**
>
> We acknowledge the challenge of providing a rigorous theoretical proof for this assumption. Therefore, our initial approach has been to establish strong empirical support. The successful reconstruction achieved by FINOLA, relying solely on local structure (first-order and linear after normalization), serves as compelling empirical evidence.
>
> Furthermore, we have empirically validated key mathematical properties associated with the model. Specifically, across multiple training runs, we have confirmed:
>
> * The invertibility of matrices $\bf{A}$ and $\bf{B}$.
> * The diagonalizability of $\mathbf{AB}^{-1}$.
>
> We believe that these empirical observations warrant further investigation and encourage the community to pursue theoretical proofs that can provide a deeper understanding of the underlying principles.
>
> ---
> $\color{blue}{[Question-2]:}$
>
> **Beyond PSNR, have you evaluated the model using perceptually motivated metrics (e.g., SSIM, FID, or IS)?**
>
> This question is also addressed in our response to Reviewer tmjQ's Question 2. Please refer to that response for a detailed discussion of FID results.
>
> To summarize, while FINOLA demonstrates strong performance in terms of PSNR, its results are less competitive with respect to FID. We attribute this discrepancy to our intentional use of the $L_2$ loss function, which was prioritized for its simplicity to emphasize that FINOLA's effectiveness does not rely on complex reconstruction loss formulations. We acknowledge that incorporating perceptual loss functions and Generative Adversarial Network (GAN) losses, as employed by models such as VQGAN, ViT_VQGAN, and Stable Diffusion, could potentially enhance FID scores.
>
> ---
> $\color{blue}{[Question-3]:}$
>
> **Is it possible to report the wave speed itself? Have you considered a framework where the wave speed varies spatially or by image rather than being invariant?**
>
> The magnitude of the wave speed (represented by the eigenvalues of $\mathbf{AB}^{-1}$) for dimension 512 is provided at
> https://tinyurl.com/2r4s8xs3. This speed value ranges from 0.5 to 1.4.
>
> We appreciate the reviewer's suggestion. While we acknowledge that reconstruction quality could potentially be enhanced by allowing the wave speed to vary spatially or across different images, this is beyond the scope of the present study. Our current investigation focuses on the scenario where wave speed remains invariant across spatial positions and images, with image variations limited to the initial condition (or compressed representation $\bf{q}$). Exploring the incorporation of spatially or image-dependent wave speeds represents a promising avenue for future research.
>
> ---
> $\color{blue}{[Question-4]:}$
>
> **Given that the derivation relies on assumptions such as the invertibility and diagonalizability of weight matrices, how sensitive is the model’s performance to these assumptions? Specifically, have you examined the condition numbers of matrices B or Q in practice, and what impact do they have on the stability and performance of the model?**
>
> While the derivation relies on the assumptions of matrix invertibility and diagonalizability, these are not explicitly enforced during model training but are validated through post-training analysis. The derivations offer a mathematical interpretation of FINOLA but do not directly affect model performance.
>
> **Empirical Validation:** We have empirically validated the invertibility of matrices $\bf{A}$ and $\bf{B}$ across multiple training runs. Furthermore, we have confirmed the diagonalizability of $\mathbf{AB}^{-1}$.
>
> **Condition Number:** The eigenvalue spectra of matrices $\bf{A}$ and $\bf{B}$, related to their condition numbers (92 and 151, respectively), are available at https://tinyurl.com/3xz2e5am.
>
> **Sensitivity Analysis:** Despite the large condition number, FINOLA is stable for the perturbations in the compressed representation space ($\bf{q}$).  To evaluate perturbation sensitivity, we performed a controlled experiment where the compressed representation ($\bf{q}$) was perturbed. Perturbed representations ($\bf{q}$$_p$) were generated by linearly interpolating between the image representation ($\bf{q}$$_i$) and and a random noise vector ($\bf{q}$$_n$):
>
> $\mathbf{q}_p=(1-p)\mathbf{q}_i+p\mathbf{q}_n$
>
> The perturbation level, $p$, ranged from 0 (no perturbation) to 1.0 (full perturbation) in steps of 0.1. FINOLA and the decoder then processed these perturbed representations to reconstruct images.
>
> The result images (see https://tinyurl.com/47whcsve) show that FINOLA is robust to small perturbations ($p$=0.1 or 0.2), but reconstruction quality diminishes with increasing perturbation.

---

> > ### Comment · Reviewer_D8KQ · 2025-04-05
> >
> > I thank the authors for their responses. I acknowledge the empirical evidence demonstrating that the model learns a traveling wave solution common among images because upon deep inspection it turns out that $A$ and $B$ are invertible and $AB^{-1}$ is diagonalizable.
> >
> > However, I have remaining concerns regarding the strong claims introduced in the paper and the provided empirical evidence. In particular, the authors claim that their method reveals inherent properties of images: that they share a particular one-way wave equation. However, this appears to be more indicative of the model selection than a property of the images. Moreover, the high PSNR and low FID scores are more indicative of blurry images that may be a result of numerical diffusion in the discretization of the one-way wave equation.
> >
> > I would be inclined to raise my score if the language regarding the inherent property of images were relaxed.

---

> > > ### Author Response · Authors · 2025-04-06
> > >
> > > Thank you for your insightful feedback regarding our claim about inherent image properties. We agree with your assessment: relaxing the claim that images inherently share a specific one-way wave equation is more accurate and better reflects the empirical nature of our findings.
> > >
> > > Therefore, we propose the following modifications to relax the claim:
> > >
> > > **Empirical Finding**: We demonstrate empirically that the FINOLA model successfully learns a highly compressed image representation ($\bf{q}$), from which images can be effectively reconstructed using a remarkably simple, spatially consistent autoregressive process (first-order and linear after normalization). This is interesting as the compressed representation captures essential image information which only relies on simple, local relationships for successful reconstruction.
> > >
> > > **Model Interpretation**: Our empirical analysis shows the learned FINOLA matrices $\bf{A}$ and $\bf{B}$ are typically invertible and $\mathbf{AB}^{-1}$ is diagonalizable, after multiple training runs. We will clarify that this allows for an interpretation where, after diagonalization, the model's dynamics resemble a set of one-way wave equations, with $\bf{q}$ as the initial condition. This is presented as an interesting perspective on the learned solution, not a proven inherent property of images themselves.
> > >
> > > **Acknowledge Theoretical Gap**: We acknowledge this wave equation perspective currently lacks rigorous theoretical proof, despite the model's strong empirical performance. We hope our work encourages further theoretical investigation.
> > >
> > > We commit to thoroughly revise the language throughout the manuscript to consistently reflect this more nuanced framing, focusing on the empirical capabilities of the FINOLA model and the interpretive nature of the wave equation connection. We believe these changes directly address your main concern.

---

### Official Review · Reviewer_vqWQ · 2025-03-17

**Overall Recommendation:** 2

**Summary:**

The authors introduce a new image encoder–decoder architecture called First Order Norm + Linear Autoregression (FINOLA). They compare it with several other image representations—including the Discrete Cosine Transform, the Discrete Wavelet Transform, and various generative models—and report that FINOLA achieves favorable results. They also test their encoder on a range of additional tasks, such as image inpainting, outpainting and self-supervised learning on Imagenet-1k dataset. However, the reviewer is not fully understood few things: (1) Image invariances (2) Line 260 contradicts the proposed method (3) the effect of perturbation on first order nonlinear differential equation

**Claims And Evidence:**

The reviewer observes a significant discrepancy between the theoretical claims made in the manuscript and their implementation in the accompanying code. Below are several specific points of concern:

Diagonalizability and Contradiction (Line 260 vs. Line 098):
The claim that 𝐴𝐵^{-1} is non-diagonalizable directly conflicts with the assumption that \Gamam is diagonal (as stated in line 102 and Equation (5)). In Figure 15, the \Gamma parameters fail to decorrelate, producing nearly identical outputs for both real and complex domains. This inconsistency undermines the theoretical foundation presented in the paper.

Impact of Small Perturbations (Equation 1):
The discussion on Equation (1) omits a key consideration: introducing a small perturbation or noise could potentially disrupt the entire generation process. Given that the model incorporates non-linearities, even minimal perturbations might significantly affect the final outputs. An analysis of the robustness of the method in the presence of noise would strengthen the manuscript.

Interpretability of Matrices
𝐴 and B:
The role and interpretability of matrices A and B remain unclear. The paper does not offer sufficient explanation or insight into how these matrices might be understood from either a theoretical or empirical perspective. Providing additional clarity—e.g., whether have specific geometric, statistical, or functional interpretations—would be beneficial.

Overall, while the paper addresses an interesting problem, the points above highlight critical gaps between the stated theory and the practical implementation. Further clarification or resolution of these issues is necessary to substantiate the claims made in the submission.

**Essential References Not Discussed:**

[1] Idempotent Generative Network, The Twelfth International Conference on Learning Representations, 2024, https://openreview.net/forum?id=XIaS66XkNA
[2] Swapping Autoencoder for Deep Image Manipulation, NeurIPS 2020.
[3] Concept Bottleneck Generative Models, ICLR 2024.

**Experimental Designs Or Analyses:**

Comparison with Recent Methods
The proposed approach is fundamentally an autoencoder, yet it does not provide any comparison with other recent techniques, such as [1]. In particular, [1, 2, 3] have been shown to achieve lower parameter counts and better FID scores than the proposed model, making a direct empirical comparison crucial for a fair evaluation.

Reconstruction Quality
While the method appears capable of capturing lower-frequency components, it struggles to preserve high-frequency details in the reconstructed images. This is especially problematic for applications in medical imaging or scenarios involving small object generation (e.g., crowds, satellite imagery), where fine-grained details are important.

Robustness to Adversarial Attacks
It remains unclear whether the proposed model’s performance holds up under various dataset/domain shifts. For instance, color or rotational shifts, as well as operations like cutmix, may significantly alter the model’s outputs. Demonstrating robustness across these perturbations would strengthen the paper’s contributions.

Evidence of Image Invariances
The discussion of image invariances is not supported by sufficiently clear experiments or evidence. Additional empirical results demonstrating these invariances under different transformations or conditions would help substantiate this claim.

References
[1] Idempotent Generative Network, The Twelfth International Conference on Learning Representations, 2024, https://openreview.net/forum?id=XIaS66XkNA
[2] Swapping Autoencoder for Deep Image Manipulation, NeurIPS 2020.
[3] Concept Bottleneck Generative Models, ICLR 2024.

**Methods And Evaluation Criteria:**

Yes, the method showcased their result on Imagenet-1k. The quantitative matrices are correct.

**Other Comments Or Suggestions:**

N/A

**Other Strengths And Weaknesses:**

The paper is well written
Dataset consider in this paper is good

**Questions For Authors:**

1. Contradiction Between Lines 260 and 098:
Does the assertion that $AB^{-1}$ is not diagonalizable conflict with the assumption that $\\Gamma$ is diagonal (line 102 and Equation (5)), especially when Figure~15 shows $\\Gamma$ failing to decorrelate and yielding nearly identical outputs for both real and complex domains?

2. Sensitivity to Small Perturbations (Equation 1):
Does the manuscript sufficiently address the impact of minor perturbations or noise on the generation process, given the reliance on non-linear transformations where even small perturbations could significantly alter the final results?

3. Role and Interpretability of Matrices $A$ and $B$:
Is there enough clarity about the purpose of matrices $A$ and $B$, including potential geometric or statistical interpretations, and how might elaborating on their function enhance the reader’s understanding of the proposed approach?

**Relation To Broader Scientific Literature:**

The work can be viewd in line of Idempotent Generative Network, The Twelfth International Conference on Learning Representations, 2024, https://openreview.net/forum?id=XIaS66XkNA, Swapping Autoencoder for Deep Image Manipulation, NeurIPS 2020., Concept Bottleneck Generative Models, ICLR 2024.

The method by nature is an image generator.

**Theoretical Claims:**

The reviewer observes significant inconsistencies between the theoretical claims and their actual implementation. Specifically:

1. Contradiction Between Lines 260 and 098:
    The manuscript asserts that $AB^{-1}$ is not diagonalizable, yet subsequently assumes that $\\Gamma$ is diagonal (line 102 and Equation (5)). Figure~15 further illustrates that the $\\Gamma$ values fail to decorrelate, yielding nearly identical outputs for both real and complex domains. This discrepancy raises questions about the validity of the stated theoretical framework.

2. Sensitivity to Small Perturbations (Equation 1):}
    The paper does not sufficiently address the potential impact of noise or minor perturbations on the generation process. Given the reliance on non-linear transformations, even small perturbations could significantly alter the final results. An in-depth robustness analysis would strengthen the manuscript.

3. Role and Interpretability of Matrices $A$ and $B$:}
    The purpose of matrices $A$ and $B$ remains unclear. The paper offers no clear explanation of how these matrices might be interpreted, either geometrically or statistically. Elucidating their function and providing interpretability would enhance the reader’s understanding of the proposed approach.

---

> ### Author Rebuttal · Authors · 2025-03-31
>
> We are grateful for the reviewer's valuable feedback.
>
> $\color{blue}{[Question-1]:}$
>
> **Contradiction Between Lines 260 and 098: Does the assertion that $\bf{AB}$$^{-1}$ is not diagonalizable conflict with the assumption that is diagonal (line 102 and Equation 5), especially when Figure~15 shows failing to decorrelate and yielding nearly identical outputs for both real and complex domains?**
>
> We appreciate the opportunity to clarify this point. The confusion between Lines 260 and 098 is resolved as follows:
>
> - Line 098: This line describes an empirical observation: "we empirically observed that $\bf{Q=AB}$$^{-1}$ matrices are diagonalizable after multiple trials of training." Subsequent derivations (Line 102 and Equation 5) are based on this observation.
>
> - Line 260: This line reiterates that the observed diagonalizability of $\bf{AB}$$^{-1}$ is a natural outcome of the training process, not an enforced constraint. Consequently, while $\bf{AB}$$^{-1}$ is not theoretically guaranteed to be diagonalizable, it consistently exhibits this property in practice across training runs.
>
> Furthermore, we wish to clarify the interpretation of Figure 15. The figure does not depict a failure to decorrelate. Instead, it illustrates two distinct variants of diagonalizable $\bf{AB}$$^{-1}$: one possessing complex eigenvalues and the other real eigenvalues. The complex eigenvalue variant demonstrates marginally superior performance (PSNR 26.1 vs. 25.1). The real eigenvalue case is achieved by constraining matrices $\bf{A}$ and $\bf{B}$ to be products of a shared real projection matrix $\bf{P}$ and distinct real diagonal matrices $\bf{H}$$_A$ and $\bf{H}$$_B$, respectively (i.e., $\bf{A=PH}$$_A$, $\bf{B=PH}$$_B$). This configuration ensures that $\mathbf{AB}^{-1}=\mathbf{P(H}_A\mathbf{H}_B^{-1})\mathbf{P}^{-1}$, where $\mathbf{H}_A\mathbf{H}_B^{-1}$ is a real diagonal matrix.
>
> ---
> $\color{blue}{[Question-2]:}$
>
> **Sensitivity to Small Perturbations (Equation 1): given the reliance on non-linear transformations where even small perturbations could significantly alter the final results?**
>
> This is an excellent point. To assess the method's sensitivity to perturbations, we conducted an experiment where perturbations were introduced into the compressed representation space. Specifically, perturbed representations ($\bf{q}$$_p$) were generated through linear interpolation between an image's representation ($\bf{q}$$_i$) and a randomly sampled noise vector ($\bf{q}$$_n$):
>
> $\mathbf{q}_p=(1-p)\mathbf{q}_i+p\mathbf{q}_n$
>
> The interpolation parameter, $p$, was varied from 0 (no perturbation) to 1.0 (full perturbation) in increments of 0.1. The perturbed vectors ($\mathbf{q}_p$) were then processed by FINOLA and the decoder to reconstruct images.
>
> The result images, available at https://tinyurl.com/47whcsve, demonstrate that FINOLA exhibits robustness to small perturbations ($p$=0.1 or $p$=0.2). However, reconstruction quality degrades as the perturbation level increases.
>
> ---
> $\color{blue}{[Question-3]:}$
>
> **Role and Interpretability of Matrices $\bf{A}$ and $\bf{B}$: including potential geometric or statistical interpretations, and how might elaborating on their function enhance the reader’s understanding of the proposed approach?**
>
> Thank you for this question! Understanding the meaning of matrices $\bf{A}$ and $\bf{B}$ is key to grasping FINOLA's core mechanism.
>
> **Definition:** Matrices $\bf{A}$ and $\bf{B}$ model the local structure of the latent vector space $\bf{z}$$(x,y)$, representing horizontal and vertical spatial transformations of the feature map:
>
> * $\Delta_x\mathbf{z}(x,y)=\mathbf{A}\hat{\mathbf{z}}(x,y)=\mathbf{P}_A\mathbf{\Lambda}_A\mathbf{P}^{-1}_A\hat{\mathbf{z}}(x,y)$
>
> * $\Delta_y\mathbf{z}(x,y)=\mathbf{B}\hat{\mathbf{z}}(x,y)=\mathbf{P}_B\mathbf{\Lambda}_B\mathbf{P}^{-1}_B\hat{\mathbf{z}}(x,y)$
>
> where $\bf{P}$$_A$ and $\bf{P}$$_A$ are the eigenvector matrices, $\bf{\Lambda}$$_A$ and $\bf{\Lambda}$$_B$ are the corresponding diagonal eigenvalue matrices. We confirm $\bf{A}$ and $\bf{B}$ full rank and diagonalizable across multiple training runs.
>
> **Interpretation:** Imagine points in the latent space connected by "springs". Matrices $\bf{A}$ and $\bf{B}$ define the stiffness of these horizontal and vertical springs, while eigenvectors ($\bf{P}$$_A$, $\bf{P}$$_B$) indicate their directions.
>
> * Diagonalization: Diagonalizing $\bf{A}$ and $\bf{B}$ simplifies this, projecting latent space so each dimension (eigenvector) is an independent spring with stiffness (eigenvalue).
>
> * Horizontal and Vertical Changes: The projected horizontal change ($\Delta_x\bf{z}$) and vertical change ($\Delta_y\bf{z}$) become scaling operations, with eigenvalues from $\bf{\Lambda}$$_A$ and $\bf{\Lambda}$$_B$ determining scaling strength along each eigenvector.
>
> In essence, $\bf{A}$ and $\bf{B}$ encode directional "stretching/compressing" factors governing local feature map changes, with eigenvalues representing the strength of these factors.

---

### Official Review · Reviewer_tmjQ · 2025-03-17

**Overall Recommendation:** 4

**Summary:**

This paper introduces an encoder-decoder framework that autoregressively reconstructs images using a first-order difference equation. The method achieves high-fidelity reconstruction, outperforms traditional encoding techniques, and is effective for self-supervised learning. The key contribution of the paper is the discovery that images share a common latent wave equation structure, offering a new perspective on image representation and reconstruction.

**Claims And Evidence:**

The claims are supported by the experiments presented in the paper; however, some experiments could be strengthened by adding additional results. For example, while the paper discusses parameter efficiency, it does not analyze training time or inference speed.

**Essential References Not Discussed:**

I am not aware of essential references not discussed.

**Experimental Designs Or Analyses:**

The experimental designs and analyses in the paper generally appear sound. The authors evaluate the proposed method (FINOLA) on  ImageNet-1K for image reconstruction and compression tasks, using standard metrics like PSNR. They also compare FINOLA against established methods like DCT, DWT, and convolutional autoencoders. However, as mentioned previously, some potential concerns include the lack of perceptual metrics (e.g., SSIM, LPIPS) in the image reconstruction evaluations, which could better capture visual quality. Additionally, the experiments don't address training time or inference speed. Finally, while ImageNet is a solid starting point, incorporating additional datasets would help strengthen the claim of robustness across different real-world scenarios. For example, datasets like COCO or ADE20K, which include diverse scenes and object categories, might help assess the method's performance in more complex, real-world settings. Additionally, datasets with different image qualities (e.g., lower resolution or noisy images) could offer insights into how well the model handles such variations.

**Methods And Evaluation Criteria:**

The proposed methods and benchmarks align well with the problem of image reconstruction, compression, and self-supervised learning. The baseline and evaluation criteria make sense for the problem at hand; however, there is a lack of perceptual metrics for image reconstruction quality. The paper primarily uses PSNR, however alternative metrics, such as SSIM or LPIPS, could provide better insight into the realism of the reconstructed images. Additionally, while ImageNet-1K is indeed a comprehensive and widely used benchmark for evaluating image reconstruction, compression, and self-supervised learning, testing on additional datasets could still provide valuable insights.

**Other Comments Or Suggestions:**

* There are some typos in the paper. For example in the abstract:
    * "empirically reveals" → "empirically reveal"
    * "to reconstruct image pixels" → "to reconstruct the image pixels"
* Reduce the size of tables (e.g. 1 to 4) and move some important results from the appendix to the main text
* It's not clear to me how to read the table in Figure 5

**Other Strengths And Weaknesses:**

The paper's strengths lie in its originality and in the experiments that demonstrate strong performance in image reconstruction while maintaining parameter efficiency, representing an improvement over existing approaches. However, the paper's readability could be improved. The flow of information is sometimes dense and mixed, making it challenging to follow. Furthermore, many important results and insights are placed in the appendix; relocating some of them to the main paper would enhance clarity and accessibility. A more structured presentation with clearer transitions between key ideas would further strengthen the work. Additionally, as previously noted, a key limitation is the lack of perceptual quality evaluation, as the paper primarily relies on PSNR. Finally, experimenting with additional datasets beyond ImageNet could help validate the method’s robustness across different domains.

**Questions For Authors:**

* Q1:  While ImageNet is a comprehensive dataset, could the authors evaluate FINOLA on other datasets (e.g., ADE20K, or medical images) to assess generalization?
* Q2: The evaluation primarily relies on PSNR, which does not fully capture perceptual quality. Have the authors considered using other metrics?
* Q3: Qualitative results are reported on a limited number of images, could the authors provide results on different images?

**Relation To Broader Scientific Literature:**

The key contributions of the paper build on prior advancements in image reconstruction, compression, and self-supervised learning by introducing a novel method for efficient image representation through multi-path networks. It extends ideas from traditional image processing techniques, improving upon them by offering better PSNR performance at smaller latent sizes. Additionally, it advances the field of self-supervised learning by showing that Masked FINOLA can compete with existing methods like MAE and SimMIM.

**Theoretical Claims:**

The paper does not provide explicit proofs for the theoretical claims made. Instead, the claims are primarily supported by experimental results, such as image reconstruction performance and comparisons with other methods like DCT, DWT, and convolutional autoencoders.

---

> ### Author Rebuttal · Authors · 2025-03-31
>
> We sincerely thank the reviewer for the thoughtful feedback and valuable suggestions, which have significantly improved the quality of our paper.
>
> $\color{blue}{\textbf{[Question 1]:}}$
>
> **While ImageNet is a comprehensive dataset, could the authors evaluate FINOLA on other datasets (e.g., ADE20K, or medical images) to assess generalization?**
>
> Thank you for suggesting the importance of assessing generalization. To address this, we conducted a novel experiment applying FINOLA to computed tomography (CT) data.
>
> **Task:** The objective of this experiment is to predict a CT image from its corresponding CT projection data. Both CT images and CT projection data are represented as 2D images.
>
> **Dataset:** The CT dataset, comprised of brain CT scans, consists of 47,000 training samples (CT image and CT projection pairs) and 6,000 test samples. The images have a resolution of 256x256.
>
> **FINOLA Implementation:** In this implementation, the CT projection data is initially encoded into a single vector, denoted as $\mathbf{q}_p$. Subsequently, two decoding branches are employed. The first branch utilizes FINOLA to generate a feature map from $\mathbf{q}_p$, followed by a decoder to reconstruct the CT projection data. The second branch linearly transforms $\mathbf{q}_p$ into a vector $\mathbf{q}_i$, which is then processed by FINOLA and a decoder to predict the corresponding CT image. Notably, the two decoding branches share the core processing components (FINOLA and the decoder) but operate on distinct, linearly correlated compressed representations ($\mathbf{q}_p$ and $\mathbf{q}_i$).
>
> **Result:** The following table presents the results of the CT image prediction task. FINOLA outperforms the two baseline methods, demonstrating its capacity to generalize to a new task and dataset.
>
> |Method| Mean Absolute Error $\downarrow$|
> |---|---|
> |InversionNet [1]|63.27|
> |SIRT[2]|45.67|
> |**FINOLA**|**31.95**|
>
> [1] InversionNet: An efficient and accurate data-driven full waveform inversion. IEEE Transactions on Computational Imaging, 2019.
>
> [2] Fast and flexible x-ray tomograph using the astra toolbox. Optics Express, 2016.
>
> ---
> $\color{blue}{\textbf{[Question 2]:}}$
>
> **The evaluation primarily relies on PSNR, which does not fully capture perceptual quality. Have the authors considered using other metrics?**
>
>
> The table below presents a comparison of FINOLA with the ***first stage*** (learning an autoencoder and vector quantization in the latent space) of multiple generative methods, assessed using both PSNR and Fréchet Inception Distance (FID) metrics.
>
> |Method|Latent Size| Channel|logit-laplace loss|$L_2$ loss|Perceptual loss|GAN loss|FID$\downarrow$|PSNR$\uparrow$|
> |---|---|---|---|---|---|---|---|---|
> | DALL-E |16x16|--|&check;||||32.0|22.8|
> | VQGAN |16x16|256|||&check;|&check;|4.98|19.9|
> | ViT-VQGAN |32x32|32|&check;|&check;|&check;|&check;|1.28|--|
> |Stable Diffustion|16x16|16|||&check;|&check;|**0.87**|24.1|
> | **FINOLA (our)**|1x1|3072||&check;|||27.8|**25.8**|
>
> While FINOLA achieves good performance in terms of PSNR, its results are less competitive with respect to FID. This divergence is a consequence of our deliberate choice to employ the $L_2$ loss function. We intentionally prioritized a straightforward loss function to emphasize that FINOLA's efficacy does not depend on complex reconstruction loss formulations. To further optimize FID scores, we acknowledge the potential benefits of incorporating perceptual loss functions and Generative Adversarial Network (GAN) losses, as demonstrated in models such as VQGAN, ViT_VQGAN, and Stable Diffusion.
>
> It is crucial to reiterate that FINOLA's primary objective differs from that of a generative model. Our focus is not on image generation per se, but rather on introducing a novel perspective for understanding images. FINOLA transforms an image into a compressed representation ($\mathbf{q}$) that enables a remarkably simple autoregressive process for image reconstruction, relying exclusively on first-order and linear (after normalization) local relationships.
>
> ---
> $\color{blue}{\textbf{[Question 3]:}}$
>
> **Qualitative results are reported on a limited number of images, could the authors provide results on different images?**
>
> Additional FINOLA reconstructed images, encompassing diverse categories such as natural scenes, human portraits, facial images, animal photographs, air and land transportation, and medical images, are available at
> https://tinyurl.com/4fdw47xm.
>
> ---
> $\color{blue}{\textbf{[Other Comments]:}}$
>
> Thanks for the helpful comments on writing and typos. We will address all these points in the final draft, including correcting the typos you noted, reducing the table sizes, moving the key results from the appendix, and improving the clarity of Figure 5.

---

> > ### Comment · Reviewer_tmjQ · 2025-04-04
> >
> > I thank the authors for their thorough rebuttal. After reading their response to my review, as well as the replies to other reviewers, I have decided to increase my score. I find the proposed approach to be novel, and the claims are well supported by extensive experimental results. That said, I am not fully confident in my assessment, as I am not familiar with the relevant literature.

---

> > > ### Author Response · Authors · 2025-04-06
> > >
> > > Thank you for your positive follow-up and for raising your score. We are pleased that our rebuttal effectively addressed your concerns. We appreciate your feedback throughout this process and confirm that the revisions discussed will be incorporated into the final manuscript.

---

### Official Review · Reviewer_Wm1J · 2025-03-25

**Overall Recommendation:** 3

**Summary:**

This paper explores the invariance in images and proposes an encoder-decoder framework based on the first-order wave equation. It works by encoding each image into an initial condition vector and then passing it to a special decoder that transforms the first-order wave equation into a linear autoregressive process to generate high-resolution feature maps and reconstruct image pixels. This approach reveals a new perspective on image understanding and provides promising directions for further exploration.

**Claims And Evidence:**

The claims in this submission are clearly and convincingly supported. The paper provides detailed experimental results and presents comparative results in the form of tables and graphs. In addition, the paper introduces new methods and techniques and tests them on multiple datasets to demonstrate their effectiveness. Therefore, it can be considered that the claims of the paper are well-founded and well-supported.

**Essential References Not Discussed:**

N/A

**Experimental Designs Or Analyses:**

The experimental design and analysis of this paper are reasonable. This paper adopts an intuitive encoder-decoder framework for image reconstruction and compares different models. In addition, this paper adjusts different hyperparameters, such as training time, number of layers, etc., to evaluate their impact on model performance. Finally, this paper also provides detailed experimental results and visualization charts to help readers better understand and evaluate the performance of the model.

**Methods And Evaluation Criteria:**

The model and evaluation criteria proposed in this paper are very reasonable for the problem and application being solved. This paper proposes a new image reconstruction framework that uses a wave equation-based method to generate high-resolution feature maps and reconstruct image pixels. At the same time, this paper also conducts extensive experimental validation of the proposed model, including comparison with other existing methods and performance analysis under different parameter settings.

**Other Comments Or Suggestions:**

N/A

**Other Strengths And Weaknesses:**

Strengths:

A new image coding method is proposed by converting the image into a one-dimensional vector and using a special decoder to generate a high-resolution feature map and reconstruct the image pixels. This method has a high reconstruction quality.

The fact that images share a set of one-way wave equations is discovered, which provides a completely new perspective to understand images and opens up the possibility for further exploration.

Weaknesses:

This method requires a lot of computing resources, especially when the feature map is high resolution. Therefore, it may be limited in practical applications.

This method is only applicable to a specific type of image, that is, images with linear structures. For other types of images, this method may not be applicable.

The explanation in the article is not detailed enough, for example, it does not go into details about how to choose the initial conditions and how to train the model. These details may affect the actual effect of this method.

**Questions For Authors:**

Please refer to the weaknesses.

**Relation To Broader Scientific Literature:**

The main contribution of this paper is to reveal the invariance in images and propose an image reconstruction method based on the one-way wave equation. This discovery is closely related to existing research in the fields of image processing and computer vision. In these fields, people have been exploring how to extract useful features and structures from raw data and how to use these features to achieve various tasks such as classification, detection and segmentation. The wave equation theory proposed in this paper provides a new perspective that can help us better understand the nature and structure of images and provide new ideas and directions for future image processing and computer vision research.

**Theoretical Claims:**

This paper proposes a new image reconstruction framework that uses a wave equation-based approach to generate high-resolution feature maps and reconstruct image pixels. Specifically, it is experimentally demonstrated that each image has a unique set of one-way wave equation solutions, and these solutions can be uniquely determined by initial conditions. This process is further transformed into a first-order norm plus linear autoregressive process, which enables efficient image reconstruction. The theoretical claims made in this paper are verified in experiments, so its theoretical correctness can be considered to be guaranteed.

---

> ### Author Rebuttal · Authors · 2025-03-31
>
> We sincerely thank the reviewer for the thoughtful feedback and valuable suggestions, which have significantly improved the quality of our paper.
>
> $\color{blue}{\textbf{[Weakness 1]:}}$
>
> **This method requires a lot of computing resources, especially when the feature map is high resolution. Therefore, it may be limited in practical applications.**
>
> While we acknowledge that generating high-resolution feature maps with FINOLA is computationally demanding, our parallel implementation enables its application in practical scenarios. As demonstrated in Figure 4, the entire process—including encoding, generating a 64x64 feature map using FINOLA, and decoding—can be completed in just 2.6 seconds on a MacBook Air equipped with an Apple M2 CPU.
>
> Furthermore, we emphasize that a key contribution of this work lies in offering a novel perspective on understanding images. FINOLA transforms an image into a compressed representation ($\mathbf{q}$) that facilitates a remarkably simple autoregressive process for image reconstruction, relying solely on local relationships that are first-order and linear after normalization.
>
> In essence, the FINOLA encoder learns a compressed representation that effectively discards explicit spatial information while preserving essential information about the image's inherent local relationships. This allows FINOLA to reconstruct the image by exploiting these consistent local structures.
>
> ---
>
> $\color{blue}{\textbf{[Weakness 2]:}}$
>
> **This method is only applicable to a specific type of image, that is, images with linear structures. For other types of images, this method may not be applicable.**
>
> Empirically, we have demonstrated the versatility of this method across a wide range of image types. Additional FINOLA reconstructed images, encompassing multiple categories such as natural scenes, human portraits, facial images, animal photographs, air and land transportation, and medical images, are available at https://tinyurl.com/4fdw47xm.
>
> Furthermore, our approach exhibits good performance with scientific images, such as seismic data and tomographic (CT) scans. For instance, in the domain of CT imaging (predicting a CT image from corresponding X-ray projection data), FINOLA not only performs well for both modalities (CT projection data and CT images) but also reveals a notable finding: the corresponding pairs of CT projection data and CT images exhibit a simple linear relationship between their compressed representations ($\mathbf{q}$). Please see more details in our reply to reviewer tmjQ (Question 1).
>
> ---
>
> $\color{blue}{\textbf{[Weakness 3]:}}$
>
> **The explanation in the article is not detailed enough, for example, it does not go into details about how to choose the initial conditions and how to train the model. These details may affect the actual effect of this method.**
>
> Thanks for the suggestion. We will add more training details. The initial condition (or compressed vector $\mathbf{q}$) is generated by the encoder. The encoder, FINOLA parameters (matrices $\mathbf{A}$ and $\mathbf{B}$), and decoder are learned end to end by using $L_2$ loss between the reconstructed and original images.
>
> Thank you for pointing out the need for more detailed explanations. Most of details are included in Appendix B.2. We will ensure clear cross-referencing between the main paper and Appendix B.2.
>
> To clarify the points you raised:
>
> - Initial Condition: The initial condition, represented by the compressed vector q, is generated by the encoder network. This network maps the input image to the compressed representation.
>
> - Training: The model is trained end-to-end using backpropagation. The encoder, FINOLA parameters (matrices $\mathbf{A}$ and $\mathbf{B}$), and the decoder network are jointly optimized to minimize the $L_2$ loss between the reconstructed and original images. Additional details are listed in Appendix B.2.

---

### Decision · Program_Chairs · 2025-05-01

**Decision:**

Accept (poster)

**Comment:**

Three reviewers out of four recommend acceptation. Despite a reminder I am not sure that the skeptical reviewer has read the authors response.

After the discussion period two reviewers have increased their score demonstrating that the rebuttal was useful in resolving the concerns that were initially raised.

In particular, the following main strengths were identified :
- this method has a high reconstruction quality,
- new perspective to understand images and opens up the possibility for further exploration,
- the approach is original
- the experiments demonstrate strong performance in image reconstruction while maintaining parameter efficiency, representing an improvement over existing approaches
- it provides empirical evidence demonstrating that the model learns a traveling wave solution common among images